# High-throughput imaging and quantitative analysis uncovers the nature of plasmid positioning by ParABS

Robin Köhler, Eugen Kaganovitch, Seán M Murray*

Max Planck Institute for Terrestrial Microbiology and LOEWE Centre for Synthetic Microbiology (SYNMIKRO), Marburg, Germany

**Abstract** The faithful segregation and inheritance of bacterial chromosomes and low-copy number plasmids requires dedicated partitioning systems. The most common of these, ParABS, consists of ParA, a DNA-binding ATPase and ParB, a protein that binds to centromeric-like *parS* sequences on the DNA cargo. The resulting nucleoprotein complexes are believed to move up a self-generated gradient of nucleoid-associated ParA. However, it remains unclear how this leads to the observed cargo positioning and dynamics. In particular, the evaluation of models of plasmid positioning has been hindered by the lack of quantitative measurements of plasmid dynamics. Here, we use high-throughput imaging, analysis and modelling to determine the dynamical nature of these systems. We find that F plasmid is actively brought to specific subcellular home positions within the cell with dynamics akin to an over-damped spring. We develop a unified stochastic model that quantitatively explains this behaviour and predicts that cells with the lowest plasmid concentration transition to oscillatory dynamics. We confirm this prediction for F plasmid as well as a distantly-related ParABS system. Our results indicate that ParABS regularly positions plasmids across the nucleoid but operates just below the threshold of an oscillatory instability, which according to our model, minimises ATP consumption. Our work also clarifies how various plasmid dynamics are achievable in a single unified stochastic model. Overall, this work uncovers the dynamical nature of plasmid positioning by ParABS and provides insights relevant for chromosome-based systems.

## Editor's evaluation

This study provides new experimental data and detailed modeling of the partitioning of low copy plasmids under the control of the ParABS system in bacteria. The dynamics of the partition complex is tracked over many generations, providing valuable data to constrain the models. The authors propose a compelling model which can manifest either regular positioning or oscillations depending on the model parameters. The research will be of interest to biologists and biophysicists interested in cellular dynamics and internal organization in bacteria.

*For correspondence:
sean.murray@synmikro.mpi-marburg.mpg.de

**Competing interest:** The authors declare that no competing interests exist.

## Introduction

To ensure that their genetic material is faithfully partitioned to daughter cells upon cell division, low-copy plasmids and bacteria employ dedicated partitioning (*par*) systems, of which ParABS is the most common (*Badrinarayanan et al., 2015*; *Kawalek et al., 2020*). The system consists of three components: (i) a centromeric-like region *parS*, (ii) a Walker-type ATPase ParA, and (iii) the protein ParB. ParA binds DNA non-specifically in its ATP-dependent dimer state and therefore coats the nucleoid. ParB dimers bind to, and spread out several kilobases from, consensus sequences within *parS* to form a condensed nucleoprotein complex, called the partition complex, that is clearly visible using

fluorescent microscopy. ParB also stimulates the ATPase activity of ParA releasing it from the nucleoid and generating a gradient of bound ParA around it. Partition complexes are believed to move up this gradient mediated by ParB-ParA bonds connecting the cargo (e.g. a plasmid or chromosomal origin) and the underlying nucleoid. In particular, the most recent molecular-scale models argue that the elastic fluctuations of the chromosome and/or ParA-ParB tethers power the directed movement of the plasmid cargo, while the self-generated ParA gradient specifies the direction (*Hu et al., 2015*; *Lim et al., 2014*).

ParABS systems fall into three main types based on their phylogeny (*Gerdes et al., 2000*). Types 1a and 1b are found on plasmids and are distinguished by their genetic organisation and the size of their genes. In particular, type 1b systems encode much smaller ParA and ParB proteins than their type 1a counterparts and have a broader host range, being found in both Gram-negative and Gram-positive bacteria. The third type is a diverse family consisting of chromosomal ParABS systems. In terms of size, they are similar to type 1a but their ParA sequences form a separate phylogenetic cluster. Perhaps the most significant distinction among the types comes from the recent result that ParB from F plasmid (a type 1a system) and several bacterial species are CTPases. ParB dimers form a DNA clamp that loads onto the DNA at *parS* sites before sliding (diffusing) along the DNA in a CTP-binding-dependent manner (*Osorio-Valeriano et al., 2019*; *Soh et al., 2019*). While this has clarified the mechanism of ParB spreading, the role of its CTPase activity in partition complex formation and positioning remains unclear. Furthermore, the much smaller ParB of the type 1b systems do not have the same CTP binding pocket, yet still confer stability to low-copy plasmids.

ParABS systems function by segregating and positioning their DNA cargo to specific positions within the cell. Typically, cargos are located symmetrically and at equally-spaced intervals across the the nucleoid i.e. at the mid, quarter or (1/6, 3/6, 5/6) positions for one, two or three cargos respectively (In the case of n cargos, their relative positions are (i-1/2)/n for i = 1, 2, …, n.). This pattern of 'home' positions is known as regular positioning and has also been observed in related ParA-like systems that position non-DNA cargo (*MacCready et al., 2018*; *Roberts et al., 2012*; *Schumacher et al., 2017*). However, the dynamics of plasmid positioning have not been characterised. So it is unclear whether the observed position distributions arise through true regular positioning in which the plasmid 'senses' the geometry of the nucleoid and positions itself accordingly (subject to stochastic variation) or through a more approximate mechanism. Indeed, both F plasmid and pB171 have been described as exhibiting oscillatory dynamics as they follow corresponding changes in the ParA gradient, which may also lead to regular positioning as a time-averaged effect (*Hatano et al., 2007*; *Ringgaard et al., 2009*; *Surovtsev et al., 2016a*). While there have been several modelling studies of plasmid positioning (*Adachi et al., 2006*; *Ietswaart et al., 2014*; *Jindal and Emberly, 2019*; *Ringgaard et al., 2009*; *Sugawara and Kaneko, 2011*; *Walter et al., 2017*) and, in particular, two recent stochastic models that incorporate the molecular mechanism of force generation (*Hu et al., 2017*; *Surovtsev et al., 2016a*), the lack of quantitative measurements of plasmid dynamics has hindered their evaluation. This is especially important as it may be challenging to distinguish noisy true positioning from approximate positioning or noisy low-amplitude oscillations.

Here, we uncover the nature of plasmid dynamics and positioning through a combination of high-throughput imaging and analysis and comparison to a minimal molecular-level computational model. We find unambiguously that the type 1a F plasmid exhibits true regular positioning as if pulled to its home positions by an over-damped spring-like force and we quantitatively reproduce its positioning and segregation behaviour in a unifying stochastic model. Furthermore, our model, an extension of the previous DNA relay model (*Surovtsev et al., 2016a*), suggests that the fraction of the nucleoid that each ParA-ATP dimer explores during its lifetime is a critical determinant of the dynamics and we confirm its prediction that single plasmids in longer cells transition to oscillatory dynamics. We also identify the ratio of the ParB-induced and the basal rate of ATP hydrolysis by ParA as a second critical model parameter. Together, these two parameters map out the entire space of plasmid dynamics and give a physical understanding of all possible dynamics including oscillations, regular positioning, static, diffusive, as well as whether the ParA distribution has a maximum or minimum at the cargo. Thus, our model, though similarly based on elastic chromosome fluctuations, unifies the existing molecular-level stochastic models (*Hu et al., 2017*; *Surovtsev et al., 2016a*) by producing all the various possible plasmid dynamics in a single model. Finally, we examine the type 1b system of pB171 and find clearer oscillatory dynamics but again dependent on the number of plasmids and cell length.

Our results show that both F plasmid and pB171 operate just below the threshold for oscillations to occur, with pB171 crossing the threshold in cells containing a single plasmid and F plasmid doing so only in cells that are additionally longer than average. Overall, our work resolves the nature of plasmid positioning and dynamics by ParABS and presents a unified stochastic model that explains the full range of behaviours in terms of well-defined system properties.

## Results

### The F plasmid is regularly positioned by a spring-like force

To clarify the nature of plasmid dynamics, we turned to a high-throughput microfluidics approach based on a custom-fabricated 'mother machine' device coupled with a segmentation, tracking and foci detection pipeline (*Figure 1A and B*). Using this approach, we tracked, at 1 min resolution, the dynamics of mini-F plasmids during many thousands of cell cycles using a fully functional ParB-mVenus fusion (*Sanchez et al., 2015*). Under our conditions, cells had a median of two ParB foci at birth and four at division (*Figure 1—figure supplement 1*) and divided approximately every 100 min. Since ParB foci separate within 5 min of plasmid replication (*Onogi et al., 2002*; *Walter et al., 2020*) and there are only a few replication events per cell cycle, in the following we will assume each ParB focus consists of a single plasmid.

Consistent with many previous works (*Adachi et al., 2006*; *Hatano et al., 2007*; *Niki and Hiraga, 1997*; *Sanchez et al., 2015*), we found that F plasmid is, irrespective of length, approximately located at mid-cell in cells with a single plasmid and close to the quarter positions in cells with two plasmids (*Figure 1C and D*). In the latter case, their positions have been more accurately specified as the quarter positions of the nucleoid (*Le Gall et al., 2016*). We found that the precision of positioning for the single plasmid case was independent of cell length, while for two plasmids it decreased weakly for cell lengths greater than 3 μm, perhaps due to variation in nucleoid segregation (*Figure 1—figure supplement 1*).

While the average position of plasmids was unambiguous, the nature of the positioning dynamics was not. In particular, it was unclear whether plasmids were consistently biased towards their average positions ('true positioning') or if they exhibited diffusive or oscillatory motion within a confined area around these positions ('approximate positioning'). Note that we are not referring here to the stochastic noisiness of positioning but rather to the nature of the positioning itself (a system with true positioning may still be noisy). As discussed above, oscillations, typically of ParA but also of the plasmid itself, have been suggested to underlie positioning in ParABS systems (*Hatano et al., 2007*; *Ringgaard et al., 2009*; *Surovtsev et al., 2016a*). In this direction, we observed, albeit very infrequently, oscillatory-like back-and-forth plasmid movements, reminiscent of some previous observations of F plasmid (*Hatano et al., 2007*). We will return to this below.

To quantitatively examine the nature of plasmid positioning, we first measured the spatial dependence of plasmid velocity (measured between two consecutive frames) as a function of long-axis position within the cell. Analysing cells containing a single plasmid, we found a clear linear dependence of the mean of the velocity on position, while its variance was constant (*Figure 1E*). Furthermore, the position and velocity autocorrelation functions showed no population level evidence of oscillatory behaviour (*Figure 1—figure supplement 2*). Rather, the velocity autocorrelation was negative at a lag equal to the sampling time, a characteristic of elastic motion. We also analysed the trajectories of cells containing two plasmids. We found a similar linear dependence of the mean velocity around the mean positions (*Figure 1F*) and no evidence of oscillatory behaviour (*Figure 1—figure supplement 2*).

These results demonstrate that F plasmid exhibits true positioning. If this was not the case, we would expect a flattening of the velocity profile around the target positions and/or evidence of oscillations in the auto-correlation of position or velocity. Altogether the observed properties are characteristic of an over-damped spring-like force, similar to that observed for the chromosomal origins of *E. coli* (*Hofmann et al., 2019*; *Kuwada et al., 2013*). Under this model, the slope of the velocity profile is the reciprocal of the characteristic timescale, $\tau$, at which elastic fluctuations act and we found this to be about 2 min. Comparable values, given the 1 min frame rate, were found by fitting to the position and velocity autocorrelation (*Figure 1—figure supplement 2A, B*). On timescales much shorter than this, plasmid dynamics are dominated by diffusion, whereas on longer timescales, the effective spring-like force dominates. As our temporal resolution is on the same order as $\tau$, we can obtain estimates,

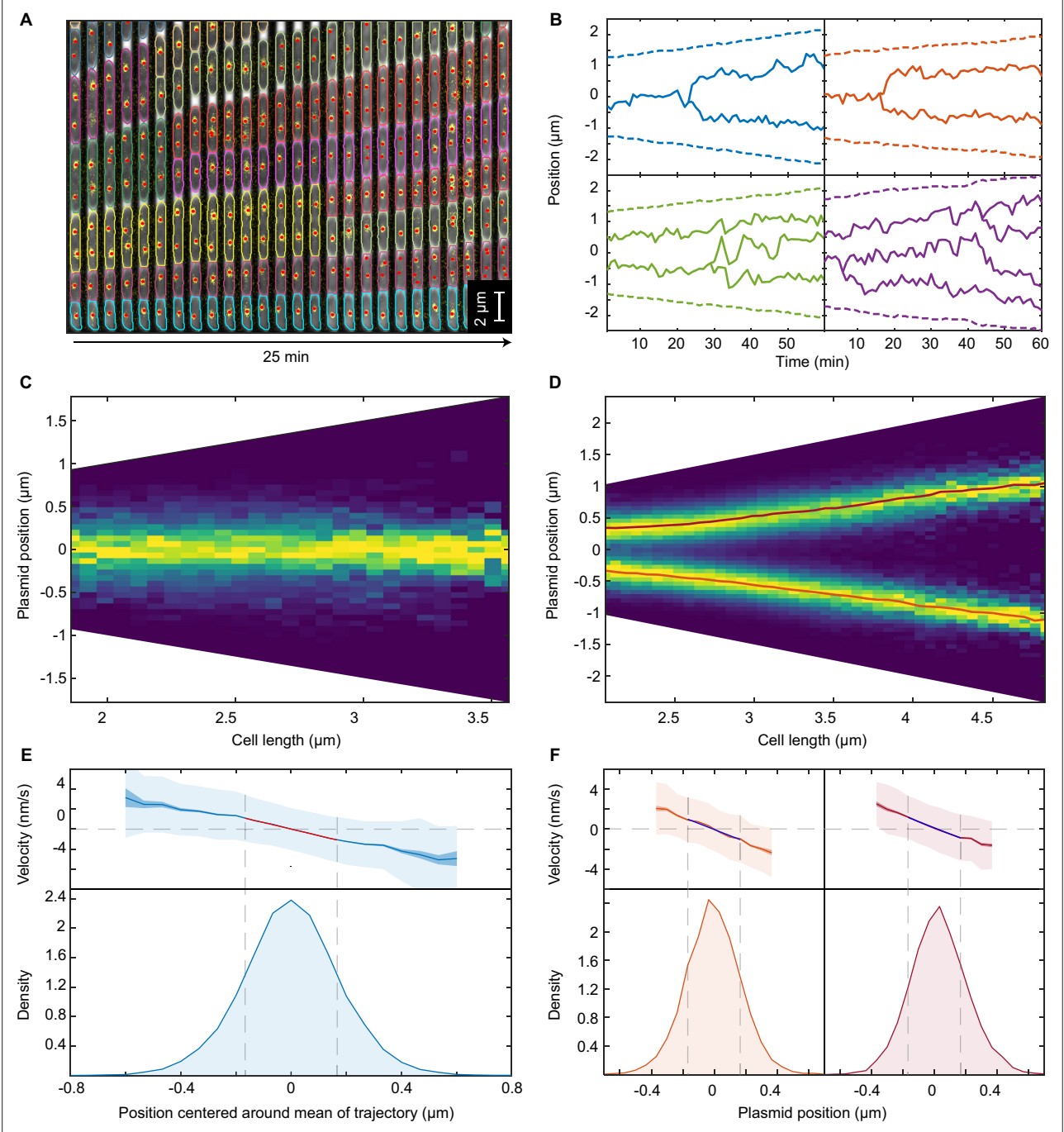

**Figure 1.** F plasmid exhibits true regularly positioning along the long axis of the cell. (**A**) Timelapse of a single mother machine growth channel (*E. coli* strain DLT3125, phase contrast overlaid with ParB-mVenus fluorescence signal). Segmentation and tracking is indicated by coloured outlines. ParB-mVenus foci are highlighted by red dots. Time interval is 1 min. (**B**) Four example trajectories of tracked ParB-mVenus foci from different cells. Dashed lines indicate cell boundaries. (**C**) Distribution of foci positions as a function of cell length in cells containing one ParB-mVenus focus. Data from 879 cell cycles. (**D**) As in (**C**) but for cells containing two ParB-mVenus foci. Data from 5044 cell cycles. Red lines indicate the position of each peak as obtained by fitting to the sum of two Gaussian functions. (**E**) Top: Mean velocity of plasmids as a function of position relative to the trajectory mean in cells containing one plasmid. The velocity is measured over two consecutive frames, taken 1 min apart. Light and dark shading indicate standard deviation and standard error respectively. The red line indicates a linear fit. Note that the standard deviation of the velocity does not depend on position. Bottom: Probability density of plasmid position relative to mean of trajectory. Standard deviation 0.182 μm. Dashed lines indicate the region used for fitting which includes at least 68.27% of all data points. (**F**) As in (**E**) but for cells containing two ParB-mVenus foci and the position is relative to the indicated lines in (**D**). Standard deviations are 0.175 μm (old pole proximal) and 0.181 μm (new pole proximal). In (**B–F**), positions and velocities are measured

*Figure 1 continued on next page*

*Figure 1 continued*

along the long cell axis. Position values are negative towards the old pole. See also *Figure 1—figure supplement 1* and *Figure 1—figure supplement 2*.

The online version of this article includes the following figure supplement(s) for figure 1:

**Figure supplement 1.** An overview of the F-plasmid.

**Figure supplement 2.** Dynamics of the F-plasmids are indicative of elastic/hindered diffusion.

under the over-damped spring model, for both the intrinsic diffusion coefficient of the plasmid D and the spring constant $k_{eff}$ of the effective force by fitting to the mean and variance of the velocity profile (see Materials and methods). We find D=(2.27±0.24) x $10^{-4}$ $\mu m^2 s^{-1}$ and $k_{eff}/(k_B T)$=36.8 ± 4.1 $\mu m^{-2}$ (bounds are the 95% confidence intervals). The latter implies a characteristic force of about 0.02 pN acting on the plasmid. Note that this estimate of the diffusion coefficient is not necessarily that of a plasmid lacking the ParABS system but rather describes the diffusive component of the dynamics in the presence of the system. To test this estimate, we tracked plasmid dynamics on a much shorter time-scale (1 s frame rate) at which diffusion is expected to dominate and measured the mean square displacement (MSD) of the plasmid. Unlike at the longer timescale, we found a linear dependence on time, and a diffusion coefficient of (2.01±0.14) x $10^{-4}$ $\mu m^2 s^{-1}$ consistent with, and in support of, the over-damped spring model (*Figure 1—figure supplement 2F, G*).

## Hopping of ParA-ATP on the nucleoid as an explanation of regular positioning

There have been two stochastic molecular-level models of plasmid positioning to date. Though different in some details, both models propose that elastic fluctuations of DNA and/or protein bonds power movement of plasmids up a gradient of DNA bound ParA-ATP. However, neither model exhibits true regular positioning as we observed for F plasmid. In the DNA relay model (*Surovtsev et al., 2016a*), plasmids oscillate across the nucleoid, reversing direction upon reaching either a pole or another plasmid, with regular positioning emerging from these oscillations only as a time-averaging effect. In the Brownian Ratchet model (*Hu et al., 2021*; *Hu et al., 2017*) on the other hand, plasmids exhibit 'local excursions' around home positions that are determined by the distance they segregate upon replication. In a narrow region of parameter space, this scheme leads to equi-positioning rather than regular positioning, that is plasmids maintain a particular inter-plasmid spacing along the long axis of the nucleoid rather than being positioned at particular locations. Given that previous coarse-grained models have displayed regular positioning (*Ietswaart et al., 2014*; *Sugawara and Kaneko, 2011*; *Walter et al., 2017*), we wondered how we could modify or extend these molecular models to exhibit true regular positioning.

Ietswaart et al. have previously shown that regular positioning can theoretically be achieved, independently of the particular mechanism of force generation, through the balancing of the diffusive fluxes of nucleoid-bound ParA-ATP into the plasmid from each longitudinal direction. If plasmids, which act as sinks for ParA-ATP, move in the direction of greatest incoming flux, then they will move toward the regularly positioned configuration since this is the unique configuration in which the diffusive fluxes balance. This 'flux balance' mechanism has since been argued to underlie positioning in several other systems (*Hofmann et al., 2019*; *Murray and Sourjik, 2017*; *Schumacher et al., 2017*). It was also realised that a critical component of the mechanism is that the ParA-ATP dimers must diffuse on the nucleoid sufficiently far before hydrolysing ATP and unbinding (*Murray and Howard, 2019*; *Subramanian and Murray, 2021*). If the associated length-scale, $s$, is too short then only ParA-ATP dimers that first bind the nucleoid close to the plasmid will have the opportunity to interact with it. As a result, the fluxes of ParA into the plasmid balance across most of the cell and the plasmid does not receive any positional information (*Figure 2A* (i-iii)). As $s$ increases the plasmid receives more positional information through the disparity in the diffusive ParA flux and beyond a threshold of half the nucleoid length true regular positioning is possible (*Figure 2A* (iv-v)). Note that the threshold decreases with the number of plasmids - with each additional plasmid, a shorter distance needs to be 'sensed' per plasmid. Sensing between plasmids occurs through competition for the same ParA-ATP dimers (*Figure 2B*). The diffusion of ParA dimers on the nucleoid referred to above could occur through hopping of dimers between DNA strands during transient unbinding events or through the

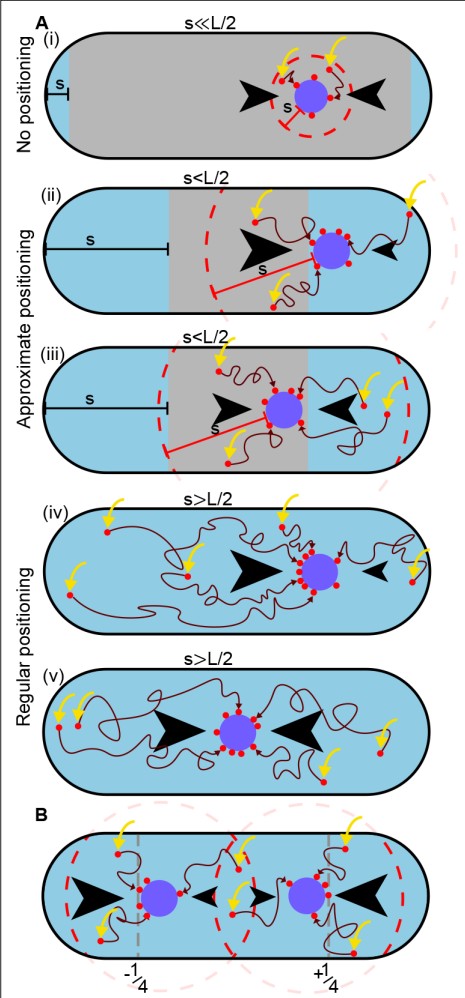

*Figure 2 continued*

plasmids but with threshold $L/(2n)$, where n is the number of plasmids. Here, sensing between plasmids occurs through competition for the same ParA-ATP dimers (the overlap between the two dashed circles). Quarter-positions are highlighted by grey dashed lines.

**Figure 2.** A difference in ParA-ATP flux can provide positional information if the diffusive length-scale is sufficiently long. The plasmid is biased away from locations at which there is a disparity in the incoming flux of ParA-ATP from either side (black arrows). However, the diffusion of ParA-ATP dimers on the nucleoid introduces a length-scale $s$, defined as the average distance dimers diffuse between association (yellow arrow) and dissociation due to hydrolysis. As a result the plasmid receives only ParA-ATP dimers that associate at most a distance $s$ from it (red dashed line). (**A**) When $s \ll L/2$ (i), where $L$ is the nucleoid length, a disparity in the fluxes into the plasmid only exists when the plasmid is very close to the poles (blue region). In the interior region (grey), the fluxes balance and the plasmid obtains no positional information. As $s$ increases (ii-iii), the region in which the plasmid receives no positional information shrinks leading to approximate mid-nucleoid positioning. When $s \gtrsim L/2$ (iv-v), ParA-ATP dimers can explore the entire nucleoid before reaching the plasmid. Hence the fluxes of ParA-ATP into the plasmid are balanced only at the mid-position. True regular positioning is achievable. (**B**) A similar argument applies to a cell with multiple

*Figure 2 continued on next page*

direct contact of DNA strands. Indeed, this has been argued to be essential for ParA gradient formation in *Caulobacter crescentus* (*Surovtsev et al., 2016b*) and was observed in vitro using single-particle microscopy (*Vecchiarelli et al., 2013*). Note that formally there still exists a non-zero disparity in the incoming fluxes into the plasmid in the low $s$ regime, however it becomes infinitesimal as $s$ decreases below the threshold (*Subramanian and Murray, 2021*).

The above argument explains why regular positioning was not observed in the DNA relay model (*Surovtsev et al., 2016a*). The key insight of that model was that bound ParA-ATP dimers experience the elastic fluctuations of the chromosomal DNA to which they bind and that these fluctuations can power the movement of the partition complex across the cell. If the partition complex has more tethers to the nucleoid in one direction, then the elastic pull of the chromosome will lead to a net force in this direction and a corresponding directed movement. However, in the model the 'home' position of each DNA-bound ParA-ATP dimer remains fixed. ParA dimers were assumed to remain bound until they interact with a (ParB-coated) plasmid that is dimers do not diffuse (hop) on the nucleoid. Hence, the diffusive length-scale, $s$, is zero and regular positioning cannot occur. On the other hand, in the Brownian Ratchet model (*Hu et al., 2017*) diffusion of ParA-ATP dimers was included but with a length-scale four times shorter than the nucleoid length. The model was therefore also not inside the regular positioning regime.

## A unifying stochastic model explains all plasmid behaviours in terms of physical parameters

Motivated by the previous discussion, we decided to develop our own minimal molecular model of ParABS positioning (*Figure 3A*). We take the DNA relay model as a starting point due to its relative simplicity (the Brownian Ratchet model explicitly models the ParA-ADP state and implements the force-dependent breakage of bonds and so has several more parameters).

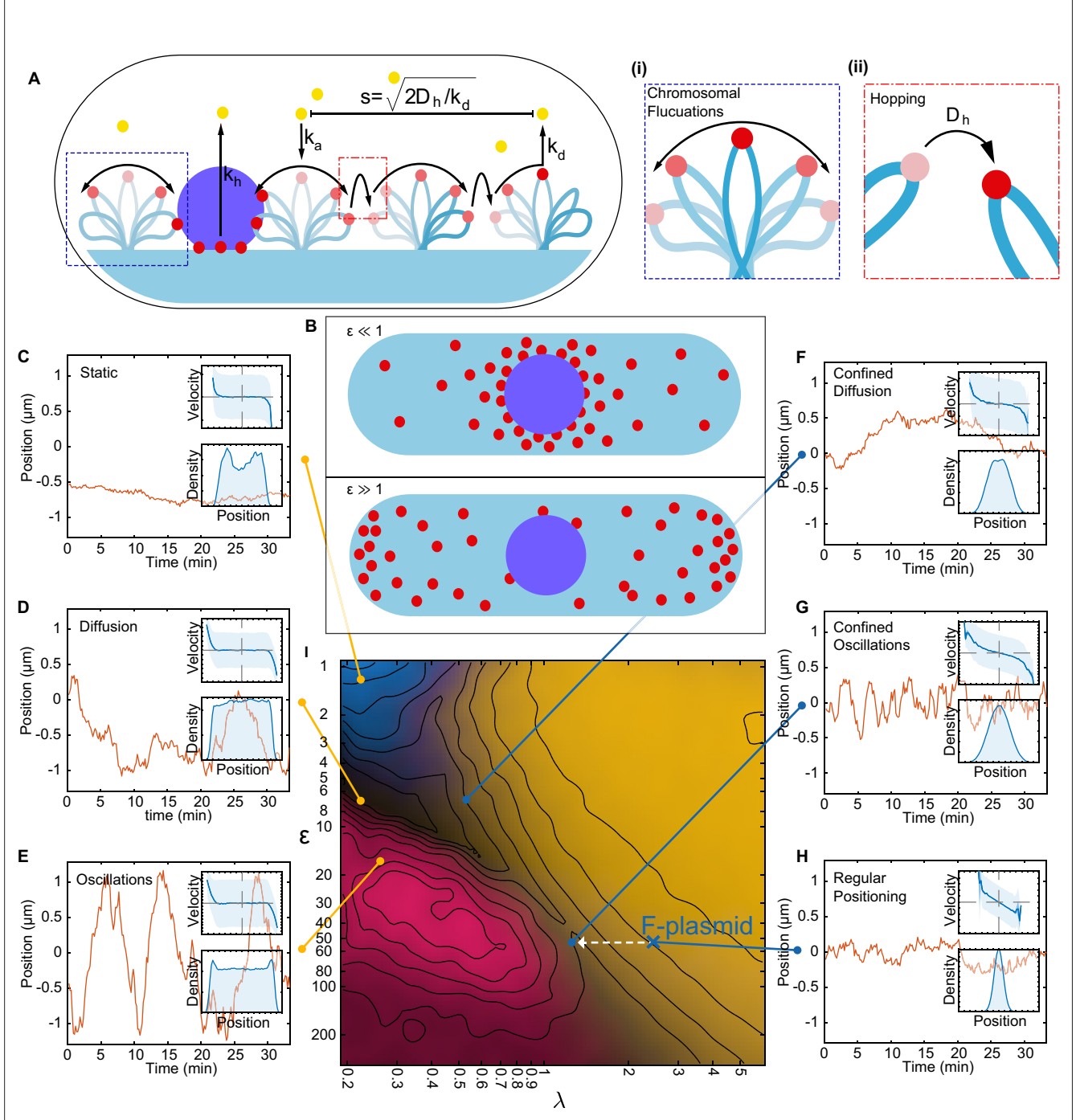

**Figure 3.** A minimal model of the ParABS system. (**A**) Schematic of the model. Light blue shading: nucleoid; light blue stroke: DNA-strand; red: nucleoid bound ParA; yellow: cytosolic ParA; purple: plasmid; arrows indicate binding and dynamics of the system; $k_a$: nucleoid binding rate of ParA; $k_d$: basal hydrolysis rate of ParA; $k_{off}$: hydrolysis rate of plasmid bound ParA. Insets: (**i**) elastic fluctuations of the chromosome, (**ii**) hopping or transfer of DNA-bound ParA-ATP dimers leads to an effective diffusion coefficient $D_h$. (**B**) A cartoon depicting low (<10) and high (>10) epsilon conditions. Low leads to a sink of ParA at the plasmid, high leads to a peak of ParA at the plasmid. (**C**) - (**H**) Example trajectories from different regimes form the phase diagram. Insets: top, velocity profile; bottom, position histogram; data from 1000 simulations. (**I**) Phase diagram obtained by varying $D_h$ and $k_{off}$. Shown in terms of the dimensionless parameters $\lambda$ and $\varepsilon$. The colour is based on an analysis of simulated trajectories as follows. Light brown: Regular positioning (confined and average position at mid-cell); blue: Static (confined and average position not at mid-cell); pink: Oscillations (highest peak in the position autocorrelation at non-zero lag); black: Diffusion (none of the previous). See Materials and methods for details. Location of the F-plasmid is marked by a cross (***Figure 3—figure supplement 1***). Number of ParA-ParB tethers and plasmid mobility can be found in (***Figure 3—figure supplement 2***).

The online version of this article includes the following figure supplement(s) for figure 3:

*Figure 3 continued on next page*

*Figure 3 continued*

**Figure supplement 1.** Fitting standard deviation of position and velocity place the F-plasmid inside the regular positioning regime.

**Figure supplement 2.** Number of ParA-plasmid tethers and their relation to plasmid mobility.

**Figure supplement 3.** The system is robust against varying the total number of ParA dimers.

**Figure supplement 4.** The effect of varying system parameters at characteristic locations in our phase diagram.

**Figure supplement 5.** 1D sweeps orthogonal to the phase diagram support the role of $\lambda$ and $\varepsilon$ in defining the dynamics.

The original DNA-relay scheme is as follows (*Surovtsev et al., 2016a*). The nucleoid is considered as a two-dimensional surface to which dimers of ParA-ATP can bind (at rate $k_a$). Upon association, dimers exhibit elastic fluctuations around their binding 'home' positions. If a dimer contacts the partition complex, itself modelled as a ParB-coated disk, it immediately binds, forming a tether between the PC and the nucleoid. The PC experiences the elastic force resulting from all attached tethers and moves as a Brownian particle under this force. Tethers are broken by ParB-induced hydrolysis (rate $k_h$), with ParA returning to a diffuse pool in the cytosol. Since the transition back to its DNA-binding competent state is slow (*Vecchiarelli et al., 2010*), the cytosolic pool of ParA-ATP is assumed to be well mixed.

Our model supplements this scheme with two additional components: diffusion of DNA-bound ParA-ATP dimers across the nucleoid (with diffusion coefficient $D_h$, where the subscript indicates diffusion of the home position) and plasmid-independent ATP hydrolysis and dissociation (with rate $k_d$). See Materials and methods for further details of the model. In the original model, dimers only unbind from the DNA due to interaction with ParB on the plasmid. However, ParA exhibits basal ATP activity (*Ah-Seng et al., 2009*). Together with diffusion on the nucleoid, plasmid-independent hydrolysis introduces a finite diffusive length-scale to the system, namely the distance a ParA dimer diffuses on the nucleoid before dissociating.

While, theoretical models with emergent behaviours are, in some sense, more than the sum of their parts, insight can be gained by identifying which physical properties of the model, typically describable by a set of dimensionless quantities, are responsible for a given behaviour. Identifying these informative quantities is critical since exploring the entire parameter space is often unfeasible. In this direction, we sought to identify the most important dimensionless quantities that characterise the behaviours of the system:

$\lambda = \frac{s}{L/2} = \frac{\sqrt{2D_h/k_d}}{L/2}$: This is the average distance, relative to half the nucleoid length, L, that each ParA-ATP dimer would theoretically diffuse on the nucleoid unhindered along each direction before unbinding due to basal ATP hydrolysis (*Figure 3A*). As discussed above, we expect that regular positioning is only possible when $\lambda \gtrsim 1$ and we confirm this below, justifying our identification of this quantity as important for the system dynamics.

$\varepsilon = \frac{k_h}{k_d}$: As $\lambda$ is the ratio of the diffusive timescale to the timescale of basal hydrolysis, we reasoned that a second quantity describing the ratio of the timescale of ParB-induced hydrolysis ($k_h$) to the timescale of basal hydrolysis would also be informative in specifying the dynamics. We expect that when this ratio, $\varepsilon$, is sufficiently large, the concentration of ParA-ATP at the plasmid will be less than that away from the plasmid (*Figure 3B*, *Figure 3—figure supplement 2*) and the opposite when $\varepsilon$ is small. This will allow us to probe the corresponding variation found experimentally.

Since the force on the plasmid is generated by the tethers between it and nucleoid-associated ParA, we reasoned that the number of nucleoid-associated ParA should also affect the dynamics of the system. Thus, we introduce a third quantity, $\theta$, the steady state number of DNA-bound ParA dimers in the absence of ParB-induced hydrolysis, given by $\theta = \frac{k_a}{k_a+k_d}n_A$, with $n_A$ being the total number of dimers in the system. Note that this involves the ratio of the third reaction rate of the system, the association rate of ParA to the nucleoid, $k_a$, relative to, once again, the basal hydrolysis rate $k_d$.

We can independently vary $\lambda$, $\varepsilon$ and $\theta$ through the parameters $D_h$, $k_h$, and $n_A$, respectively. However, we found that while $n_A$ had, unsurprisingly, a strong effect on the degree of stochasticity in the system, it had little effect on the nature of the dynamics (*Figure 3—figure supplement 3*). The different regimes were clearly detectable from at least $n_A = 50$ dimers. We therefore focused on $\lambda$ and $\varepsilon$.

We first considered the case of a single plasmid and performed simulations of the model over a range of values of these two quantities. The other parameters were fixed at estimated values (see

**Table 1.** Model Parameters.

| Parameter | Brief description | Value | Source |
|---|---|---|---|
| $k_a$ | Association rate to the nucleoid of cytosolic ParA | 0.19 s⁻¹ | As *Surovtsev et al., 2016a*. Based on in vitro measurement from *Vecchiarelli et al., 2010*. Results in 95% ParA nucleoid association in the absence of a plasmid. |
| $k_d$ | Dissociation due to basal hydrolysis rate of ParA | 0.01 s⁻¹ | Based on in vitro measurement from *Hwang et al., 2013*; *Vecchiarelli et al., 2013*. |
| $k_h$ | Tether dissociation due to plasmid stimulated hydrolysis of ParA | 0.01–3 s⁻¹ | Sweeped over in this study. |
| $D_p$ | Diffusion coefficient of the plasmid | 3x10⁻³ μm²s⁻¹ | As *Surovtsev et al., 2016a*. Based on MSD of a Δ*par* plasmid. |
| $D_h$ | Diffusion coefficient of ParA home position on the nucleoid | 3.22x10⁻⁴ to 0.29 μm²s⁻¹ | Sweeped over in this study. |
| $D_A$ | Diffusion coefficient of DNA-bound ParA due to chromosomal fluctuations | 0.01 μm²s⁻¹ | As *Surovtsev et al., 2016a*; Based on *Javer et al., 2014* and *Weber et al., 2010*. |
| $W$ | Width of the cell | 0.95 μm | This study |
| $L$ | Length of the cell | 2.5–4.34 μm | This study |
| $dt$ | Simulation time step | 0.001 s | This study |
| $R_p$ | Radius Plasmid | 0.05 μm | As *Surovtsev et al., 2016a*. Estimate from *Sanchez et al., 2015*. |
| $R_A$ | Radius ParA | 0.002 μm | As *Surovtsev et al., 2016a*. Based on ParA crystal structure from *Leonard et al., 2005*. |
| $\sigma_x$ | Width of elastic fluctuations of the chromosome along long cell axis | 0.1 μm | As *Surovtsev et al., 2016a*. |
| $\sigma_y$ | Width of elastic fluctuations of the chromosome along short cell axis | 0.05 μm | As *Surovtsev et al., 2016a*. |
| $n_A$ | Number of ParA dimers | 500 | Midrange estimate from *Adachi et al., 2006*; *Bouet et al., 2005*; *Lim et al., 2014*. |
| $n_p$ | Number of Plasmids | 1–5 | This study |

*Table 1*) and the length of the simulated nucleoid was chosen to match the average length of cells with one F plasmid (*Figure 1—figure supplement 1*). These simulations produced a range of plasmid behaviours with a clear dependence on the position in this $(\lambda, \varepsilon)$ phase space (*Figure 3C–I*). In particular we observed an interface at approximately In particular we observed an interface at approximately $\lambda \approx 1$ separating two regimes at small $\lambda$, with a single regime at large $\lambda$.

We first consider small $\lambda$. The plasmid was found to move diffusively for $\varepsilon < 10$ as evidenced by its zero mean velocity across the nucleoid and flat-topped position distribution. However, the diffusivity of the plasmid decreases with decreasing $\varepsilon$ so that at the lowest values of $\varepsilon$ studied, at which the hydrolysis rate $k_h$ at the plasmid is comparable to the rate $k_d$ away from it, the plasmid moves so slowly that it is effectively static on the timescale of our simulations (35 min) and remains approximately at its initial position.

When $\varepsilon$ is increased beyond 10 (i.e. when the hydrolysis rate at the plasmid is much greater than that away from it), we observed clear oscillatory behaviour, similar to that observed in previous models. This transition from static to diffusive to oscillatory can be understood in the terms of the differing timescale of tether dissociation on the one hand and the ParA repletion rate on the other (*Hu et al., 2015*; *Walter et al., 2017*). In the oscillatory regime, tethers break (due to ATP hydrolysis) faster than they can be replaced. This leads to a ParA depletion zone behind the plasmid that reinforces its movement in the same direction. The result is directed motion until the nucleoid edge, at which point the depletion zone fills, causing the plasmid to change direction. This turnaround time is apparent in the boundary peaks in the position distribution (*Figure 3E*). At even higher values of $\varepsilon$, tethers are so short lived that the dynamics become once again diffusive.

At the interface region $\lambda \approx 1$, the plasmid becomes confined to the centre region of the nucleoid where it exhibits either diffusive or oscillatory motion depending on the position along the interface.

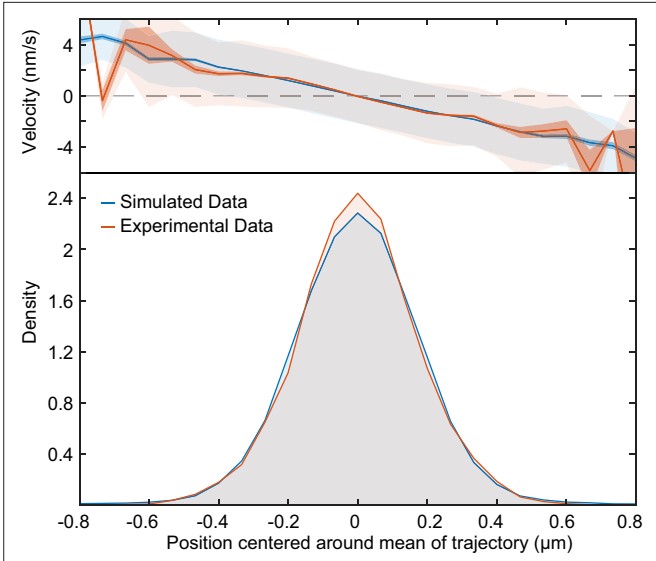

**Figure 4.** Fitted position of the F-Plasmid. Simulated data (blue) from **Figure 3H** at $(\lambda, \varepsilon) = (2.66, 56.42)$ compared to the experimental data (red) from **Figure 1E**. Top: Mean velocity of plasmids as a function of position relative to the trajectory mean. Light and dark shading indicate standard deviation and standard error respectively. Bottom: Probability density of plasmid position relative to mean of trajectory.

As $\lambda$ is increased further, the positioning becomes more precise, the confined region shrinks and the plasmid exhibits true regular positioning. This is consistent with our hypothesis of the importance of the diffusive length-scale for the functioning of the flux-balance mechanism. Within this large regular positioning regime the position distribution, velocity profile and autocorrelation (**Figure 3—figure supplement 1**) have qualitatively the same form as we observed experimentally for cells containing a single plasmid and we found excellent quantitative agreement for $(\lambda, \varepsilon) = (2.66, 56.42)$ (**Figure 4** and **Figure 3H**, blue cross in **Figure 3I**). Interestingly, these parameter values suggest that while the dynamics of a single F plasmid sits within the regular positioning regime, it is not far from the interfacial region of confined oscillations.

We next measured how many ParA tethers were associated with the plasmid as the parameters were varied. We found the numbers of tethers varies positively with $\lambda$ and negatively with $\varepsilon$, consistent with an increase in the flux of ParA dimers into the plasmid and longer tether lifetimes respectively (**Figure 3—figure supplement 2A**). Interestingly we found a clear relationship between the mobility of the plasmid and the number of ParA tethers, with the oscillatory regime having the fewest number of (simultaneous) tethers and the greatest mobility and the regularly-positioning regime at low $\varepsilon$, having the most tethers and the slowest movement (**Figure 3—figure supplement 2B**). This was also apparent from kymographs of the ParA distribution (**Figure 3—figure supplement 2C**). Note however, that the latter regime does exhibit regular positioning - it simply takes a very long time for the plasmid to move to mid-position. Similarly, in the 'static' regime the plasmid actually exhibits very slow diffusive motion. In this sense, there are really only three regimes (diffusive, oscillatory and regular positioning) and their interfaces.

Finally, we explored how the other parameters of the model affect the dynamics. We varied the main parameters across four orders of magnitude centred on the set used in **Figure 4** (which lies in the regular positioning regime). We found that only the two parameters varied in our sweep were able to push the system into the static or oscillatory regimes (**Figure 3—figure supplement 4**). Starting from the diffusive regime, this could also be achieved by changing the basal hydrolysis rate $k_d$ consistent with how the dimensionless quantities $\lambda$ and $\varepsilon$ depend on it (changing $k_d$ should move the system diagonally in the phase diagram). To confirm the role of these dimensionless quantities in determining the dynamics, we varied $D_h$, $k_d$, $k_h$ and $k_a$ simultaneously over two orders of magnitude. This has the effect of modulating the turnover rate of ParA tethers while keeping $\lambda$, $\varepsilon$ and $\theta$ fixed. We found no change in the nature of the dynamics beyond an expected increase in the frequency of the fluctuations in the plasmid position as the tether turnover rate is increased (**Figure 3—figure supplement 5**).

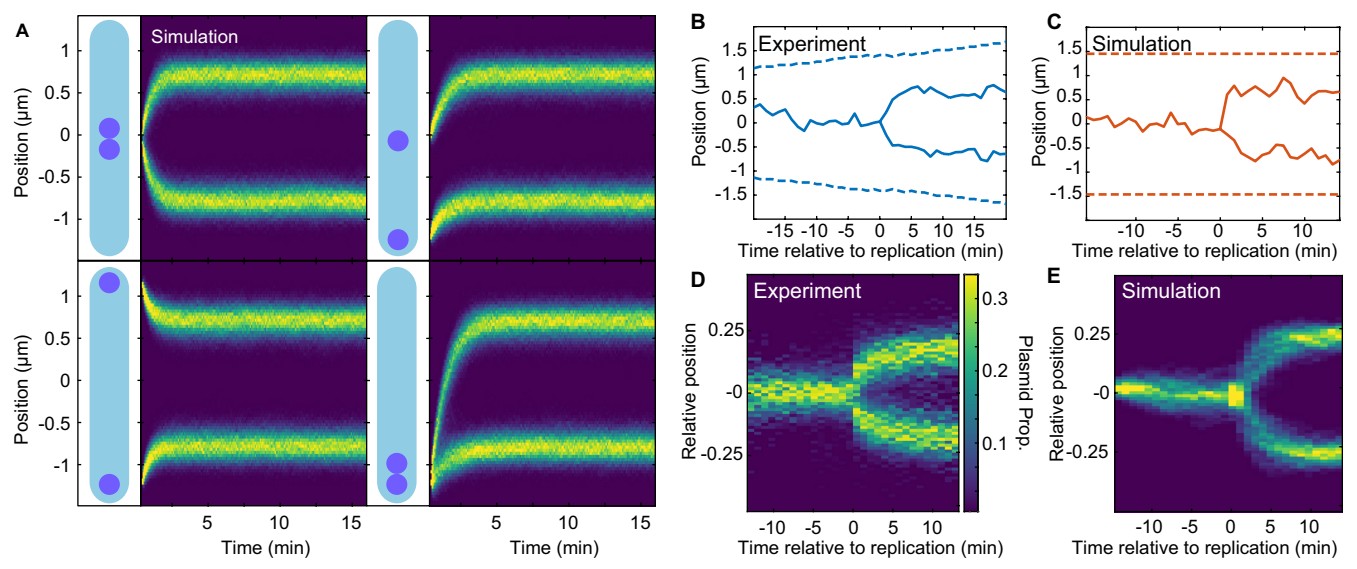

**Figure 5.** Regular positioning of two simulated plasmids. (**A**) Kymographs showing the distribution of plasmid positions starting from different initial positions along the long axis. Data is from 1000 simulations each. Nucleoid length is chosen to match our experimental data (see *Table 2*). (**B**) Example of F plasmid replication (splitting ParB-mVenus focus) event. (**C**) Example simulated replication event. (**D**) Kymograph of F plasmid splitting events as in (**B**). Data from 500 cell cycles were combined according to the time of focus splitting. (**E**) Kymograph of simulated plasmid replication. Upon replication, both plasmids occupy the same position but only one inherits the ParA-ParB tethers. This does not affect the result as the number of tethers equilibrates rapidly. Data from 1000 simulations. Note that in (**D**) position is relative to cell length, rather than nucleoid length as in the simulations (**D**). See also *Figure 5—figure supplement 1A*.

The online version of this article includes the following figure supplement(s) for figure 5:

**Figure supplement 1.** Regular positioning with multiple plasmids.

We also explored if regular positioning is achievable in the absence of ParA-ATP diffusion on the nucleoid (i.e. $D_h = 0$). However, we found that it only occurs if the length scale of chromosome fluctuations $\sigma_{x,y}$ is increased far beyond its measured value of about 0.1 µm to 1 µm (*Figure 3—figure supplement 4A*). At this unphysically high value, each DNA-bound ParA dimer can, through the fluctuations of the underlying DNA, interact with the plasmid over long distances and from across the cell. The plasmid is therefore positioned at mid-cell because this is the only position where the net force from all ParA dimers balances. However, based on the measurements of the chromosome fluctuations (*Lim et al., 2014*; *Surovtsev et al., 2016a*; *Wiggins et al., 2010*), we believe this regime is not biologically relevant.

## Regular positioning of two plasmids

We next considered the case of cells having two and more plasmids. We found that our model could reproduce the same quarter positioning as observed for F plasmid (*Figure 5A*). Importantly, regular positioning was achieved irrespective of where the two plasmids were initially positioned. This is in contrast to the model of Hu et al., in which plasmids move apart a fixed distance. We also simulated plasmid replication by duplicating one plasmid during the simulation. We found that the replicated plasmids moved apart rapidly towards the quarter positions in a qualitatively similar way as we observed in our experimental data (*Figure 5B–E*). We expect that better knowledge of the biochemical parameters would further improve this comparison.

Examining the phase diagram for more than one plasmid, we found the boundary of the regularly positioning regime expands to lower $\lambda$ values (*Figure 5—figure supplement 1B*). This is consistent with $s$, the distance ParA-ATP dimers diffuse on the nucleoid, needing to be greater than $\frac{L}{2n}$ for regular positioning to occur (*Figure 2*). When we displayed the phase diagrams in terms of $\lambda_n = n\lambda = \frac{s}{L/2n}$, we found that they all collapsed onto each other, with regularly positioning only occurring for $\lambda_n \gtrsim 1$, further confirming the importance of this parameter (*Figure 5—figure supplement 1C*).

**Table 2.** Simulation parameters used in figures.

| Figure | $k_h$ $(1/s)$ | $D_h$ $\left(\mu m^2/s\right)$ | $L$ $(\mu m)$ | $n_p$ |
|---|---|---|---|---|
| *Figure 3C*, *Figure 3—figure supplement 1B*, *Figure 3—figure supplement 4A*, *Figure 3—figure supplement 5B* | 0.0133 | 0.000440 | 2.53 | 1 |
| *Figure 3D*, *Figure 3—figure supplement 1C*, *Figure 3—figure supplement 4B*, *Figure 3—figure supplement 5B* | 0.0769 | 0.000440 | 2.53 | 1 |
| *Figure 3E*, *Figure 3—figure supplement 1D*, *Figure 3—figure supplement 4C*, *Figure 3—figure supplement 5B* | 0.1785 | 0.000623 | 2.53 | 1 |
| *Figure 3F*, *Figure 3—figure supplement 1E* | 0.0752 | 0.002497 | 2.53 | 1 |
| *Figure 3G*, *Figure 3—figure supplement 1F* | 0.5642 | 0.014162 | 2.53 | 1 |
| *Figure 3H*, *Figure 3—figure supplement 1G*, *Figure 3—figure supplement 4D*, *Figure 3—figure supplement 5B* | 0.5642 | 0.056760 | 2.53 | 1 |
| *Figure 3I*, *Figure 8*, *Figure 3—figure supplement 1A*, *Figure 3—figure supplement 2*, *Figure 3—figure supplement 3\**, *Figure 3—figure supplement 4E* | 0.01–3 | 0.000322–0.29 | 2.53 | 1 |
| *Figure 4* | 0.5642 | 0.056760 | 2.53 | 1 |
| *Figure 5A* | 0.5642 | 0.056760 | 2.91 | 2 |
| *Figure 5C, E†* | 0.5642 | 0.007179 | 2.91 | 1 ->2 |
| *Figure 6A* (inset: orange, blue) | 0.5642 | 0.056760 | 1.82,4.93 | 1 |
| *Figure 5—figure supplement 1A* | 0.5642 | 0.056760 | 3.54 | 1 ->5 |
| *Figure 5—figure supplement 1B, C* | 0.01–3 | 0.000322–0.29 | 2.53, 2.91, 3.67, 4.34 | 1,2,3,4 |

*$n_A$ was changed in the range of 5 to 1000.

†kd was changed to 0.001.

## Length dependent transition to the confined oscillatory regime

While F plasmid operates within the regular positioning regime, our model predicts that its dynamics can become oscillatory by decreasing $\lambda$ (*Figure 3I*, white arrow). Since $\lambda$ depends inversely on the nucleoid length, L, we wondered whether oscillations would appear in longer cells. When we used the same model parameters determined above but with different lengths, we found that the system could indeed enter the (confined) oscillatory regime, commensurate with the length-induced decrease in $\lambda$ (*Figure 6A*).

Motivated by these results, we went back to our F plasmid data and examined cells harbouring one plasmid with greater than average cell length. Consistent with our simulations, we found multiple cell cycles in which the plasmid was initially stably positioned at mid-cell but as the length of the cell increased, appeared to display low-amplitude oscillations (*Figure 6B*). To investigate if this transition was reproducible, we developed a method to classify segments of trajectories as oscillatory (or processive), regularly positioned or undetermined based on the velocity autocorrelation between consecutive frames (*Figure 6—figure supplement 1*). Binning the individual time-points from these classified segments according to cell length revealed the relative abundance of the two populations (*Figure 6C*). We found a marked increase in the proportion of oscillatory segments from cell lengths of about 3 μm, with up to 50% of timepoints being classified as oscillatory, consistent with our prediction. This also confirms a previous rough estimate that F plasmid operates not far below the threshold of oscillatory instability (*Walter et al., 2017*).

Interestingly, the same analysis on cells containing two plasmids revealed a significantly smaller proportion of oscillatory segments and a weaker length dependence. However, this is again consistent with our prediction that, in a cell with $n$ plasmids, ParA-ATP dimers need to diffuse on the nucleoid an average distance of at least $L/(2n)$ in order for the plasmids to sense each other and regular positioning to occur (*Figure 2B*). Since, within the population, this threshold distance is greatest for cells containing a single plasmid (*Figure 6D*), it is in these cells that we are most likely to observe a transition to oscillatory behaviour. More specifically, these results suggest that ParA-ATP dimers diffuse

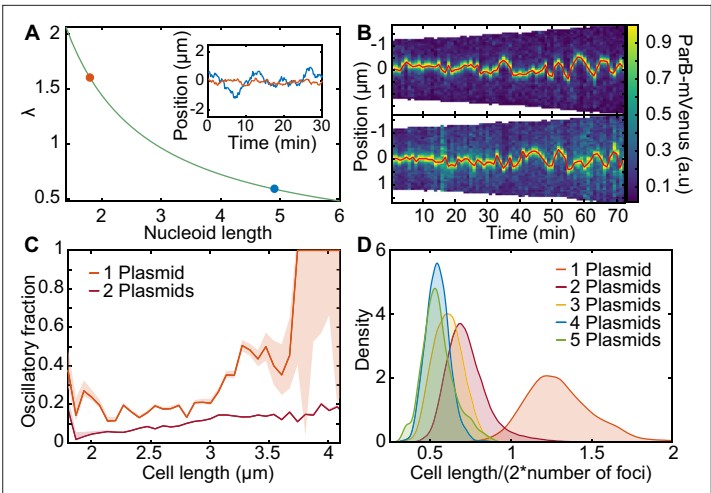

**Figure 6.** Length-dependent effect on the transition from regular positioning to oscillations. (**A**) inverse dependence of $\lambda$ on nucleoid length. inset: simulated trajectories at the highlighted lengths/$\lambda$ values. (**B**) kymographs of parb-mvenus signal along the long axis of two cells depicting the transition from regular position to low-amplitude oscillations. the plasmid trajectory is highlighted in red. (**C**) the fraction of the oscillating population plotted against cell length for cells with one (879 cell cycles) or two plasmids (5044 cell cycles) (see *figure 6—figure supplement 1*). (**D**) the distribution of threshold length-scale (l/2n) for cells containing different numbers of plasmids. data from 16,346 cell cycles.

The online version of this article includes the following figure supplement(s) for figure 6:

**Figure supplement 1.** Classification of segments of trajectories.

**Figure supplement 2.** Oscillatory behaviour decreases upon plasmid replication.

**Figure supplement 3.** There is no significant inheritance of oscillatory behaviour.

a distance of about 1.5 µm before dissociating. We also examined how oscillations are affected by changes in plasmid number within individual cells i.e. upon plasmid replication (*Figure 6—figure supplement 2*). We found that oscillatory behaviour appeared to decrease in that a classification of oscillatory dynamics before replication was not a reliable indicator of oscillatory dynamics afterwards. The same was true across generations - we observed a rapid decay in the autocorrelation for containing an oscillatory trajectory segment (*Figure 6—figure supplement 3*).

## pB171 operates closer to the oscillatory regime than F Plasmid

We have shown above that F plasmid is, for the most part, regularly positioned within cells, with a transition towards oscillatory behaviour only occurring in those cells with greatest sensing threshold $L/\left(2n\right)$, that is in cells with the lowest plasmid concentration. Might other ParABS systems exhibit more pronounced oscillatory dynamics? To explore this, we examined the dynamics induced by the ParABS system of the plasmid pB171. We chose this system as it has previously been described as oscillatory (*Ringgaard et al., 2009*) and it belongs to the other family of ParABS systems, namely type 1b (F plasmid is type 1a).

Using a previously constructed TetR/*tetO* labelling system, we first determined the copy number of this system and found it to be comparable to F plasmid (*Figure 7—figure supplement 1*). We then examined plasmid dynamics in cells containing a single plasmid and found clear unambiguous oscillatory behaviour in ~80% of such cells (*Figure 7A*), in stark contrast to F plasmid (*Figure 7—figure supplement 2*). This was reflected in the flat-topped plasmid position distribution (*Figure 7B*), which was very different from that of F plasmid and more similar to what we obtained in the oscillatory regime of our model (*Figure 3—figure supplement 1*). More importantly, the oscillatory nature of the dynamics was reflected in the position and velocity autocorrelations (*Figure 7C and D*), including a positive velocity autocorrelation between consecutive frames, which is a signature of processive motion (*Figure 1—figure supplement 2*). These curves qualitatively matched those obtained in our model within the oscillatory regime (*Figure 3—figure supplement 1*).

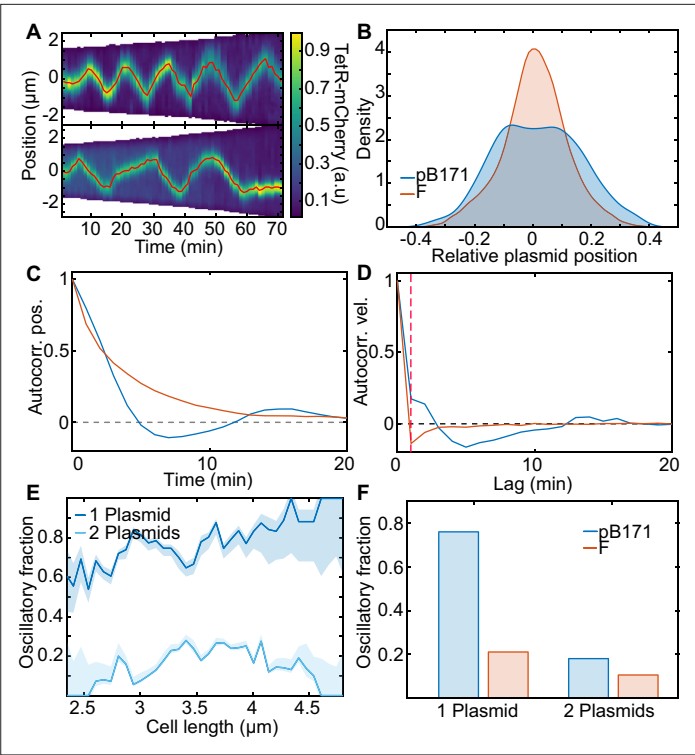

**Figure 7.** pB171 and its differences to F. (**A**) Two kymographs (pB171) of TetR-mCherry signal along the long axis of two cells with one plasmid. Red line indicates the trajectory of the plasmid (more examples of trajectories can be found in *Figure 7—figure supplement 2*). (**B**) Relative position position distribution for pB171 (blue, 68 cell cycles) and F-Plasmid (red, 879 cell cycles) for cells with one plasmid. (**C,D**) Position and velocity autocorrelation for pB171 (blue) and F plasmid (red). Positive velocity autocorrelation at 1 min (dashed red line) indicates processive dynamics (see *Figure 1—figure supplement 2*). (**E**) The proportion of trajectory time points classified as oscillatory from pB171 plotted against cell length for cells containing one or two plasmids. Data from 68 and 117 cell cycles, respectively. (**F**) Comparison between oscillating population of pB171 and F-Plasmid with one and two plasmids on whole population level.

The online version of this article includes the following figure supplement(s) for figure 7:

**Figure supplement 1.** Distribution of copy number of pB171 as measured by the number of fluorescent foci within cells.

**Figure supplement 2.** Comparison of pB171 and F-plasmid.

**Figure supplement 3.** Comparison of pB171 and F-plasmid in cells containing two plasmids.

---

We also found that oscillatory dynamics were more likely in longer cells (*Figure 7E*), consistent with our prediction of the importance of nucleoid length in determining the dynamical regime (through $\lambda$). Overall oscillations were almost four times as likely for pB171 as for F plasmid (*Figure 7F*). However, this was much reduced when we considered cells with two plasmids, for which oscillations were much less apparent (*Figure 7E and F*, *Figure 7—figure supplement 3*). This suggests that, similar to F plasmid, the ParABS system of pB171 does not lie entirely within the oscillatory regime, but only enters it for cells containing a single plasmid, in which the sensing distance required for regular positioning is longest (see above).

## Discussion

ParABS systems have become a paradigm of self-organisation within bacterial cells. Yet, it is still unclear how these systems function. Three main research questions can be identified: (1) How does ParB spread over the centromeric region to form the nucleoprotein partition complex (PC), (2) What is the nature of the force underlying directed movement of the PC, and (3) How is the directionality and positioning of the PC specified? The recent discovery (*Jalal et al., 2020*; *Osorio-Valeriano et al.,*

*2019*; *Soh et al., 2019*) that ParB dimers are CTP-dependent DNA clamps that load onto, and slide away from, *parS* sites has shed light on the first question, at least for the type 1a and chromosomal ParABS systems. While the force-generating mechanism underlying directed movement has yet to be definitively identified, the current proposal, supported by modelling, is that movement is powered by the elastic fluctuations of the chromosome and/or ParA-ParB tethers (*Hu et al., 2015*; *Lim et al., 2014*).

The nature of partition complex positioning on the other hand has yet to be resolved. On the experimental side, quantitative measurements of plasmid dynamics have been lacking. It has therefore not been possible to determine if plasmids are truly regularly positioned, which implies some method of geometry sensing or if they are rather recruited by, for example, regions of higher DNA density (*Le Gall et al., 2016*). Furthermore, while oscillatory dynamics have been observed, it was not clear whether this is representative of how these systems operate. On the modelling side, there have been several studies, both deterministic (*Adachi et al., 2006*; *Ietswaart et al., 2014*; *Jindal and Emberly, 2019*; *Sugawara and Kaneko, 2011*; *Walter et al., 2017*) and stochastic (*Hu et al., 2021*; *Hu et al., 2017*; *Ietswaart et al., 2014*; *Surovtsev et al., 2016a*), each producing some set of dynamical behaviours. However, the lack of quantitative dynamical measurements has meant that none of these models has been quantitatively compared or tested against experimental observations. It has therefore been unclear which model or, more specifically, which model ingredients, best describe plasmid positioning by ParABS.

In this work, we addressed this deficiency by first performing a high-throughput quantitative analysis of plasmid dynamics. We determined that the motion of F plasmid, which hosts a type 1a ParABS system, is consistently biased towards specific home positions (mid-cell in cells containing a single plasmid, approximate quarter positions in cells with two plasmids) as if pulled by a spring-like force. The precision of this 'regular positioning' was seen in the consistently spatially varying average velocity of the plasmid and supports the presence of a geometry sensing mechanism. However, such

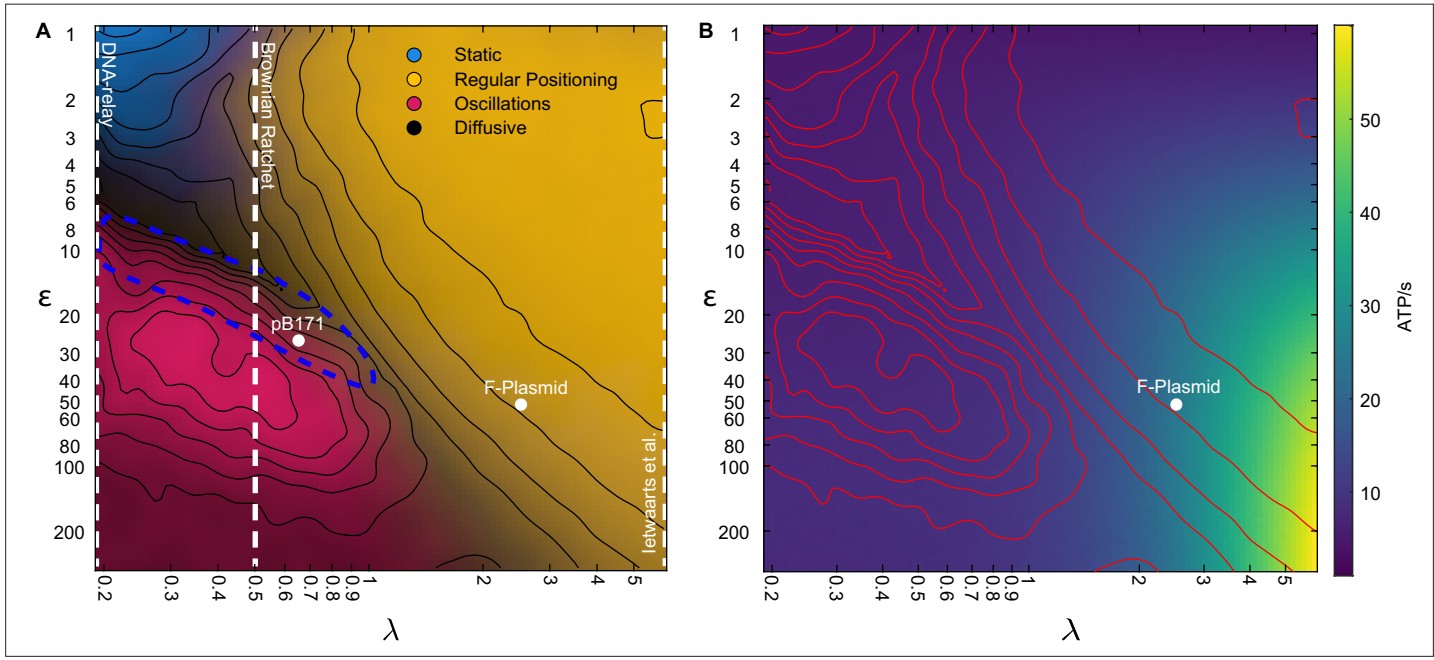

**Figure 8.** The model encompases existing stochastic models as limiting cases. (**A**) The phase diagram for a single plasmid from *Figure 3I* with the conceptual location of existing stochastic models indicated. In the DNA relay model, bound ParA-ATP dimers do not diffuse on the nucleoid and so $\lambda = 0$. The model cannot produce regular positioning. In the model of Ietswaart et al., ParA dimers diffuse on nucleoid but only dissociate by interacting with the plasmid, therefore $\lambda = \infty$ and the model lies entirely in the regular positioning regime. For both models, the y-axis represents $k_h$ the hydrolysis rate at the plasmid. The Brownian Ratchet lies between these two extremes. The length scale associated to ParA diffusion is finite but its value was fixed at $s = 0.5$ μm (given $\lambda = 0.5$ for a 2 μm nucleoid), so that the regular positioning regime is not explored. The locations of the ParABS systems of F plasmid and pB171 in cells containing a single plasmid are shown. The location of pB171 is an estimate based on a qualitative comparison of its dynamics. The blue dashed line marks the region in which the period of the oscillations resembles the experimental observations. (**B**) ATP consumption rate. Red lines are the contours from (**A**).

**Table 3.** Model comparison.

| | Hopping and relay (This study) | DNA-Relay (Surovtsev et al., 2016a) | Brownian-Ratchet (Hu et al., 2017) | Model of Ietswaart et al., 2014 | Model of Schumacher et al., 2017† |
|---|---|---|---|---|---|
| Elastic fluctuations | ✓ | ✓ | ✓ | X | ✓ |
| Basal-hydrolysis | ✓ | X | ✓ | X | X |
| ParA diffusion on nucleoid | ✓ | X | ✓ | ✓ | ✓ |
| Length-scale of ParA diffusion on the nucleoid | finite | 0 | finite | infinite | infinite |
| ParA diffusion on partition complex | X | X | X | X | ✓ |
| Hydrolyzed nucleoid bound ParA state | X | X | ✓ | X | X |
| Limited binding of ParA to partition complex | X | X | ✓ | ✓ | X |
| Limit on tether length | X | X | ✓ | X | X |
| Cytosolic ParA pool | Well mixed | Well mixed | Unlimited pool, well mixed | Well mixed | Well mixed |
| Observed behaviour | Diffusion, regular positioning, static, oscillations | Oscillations | Diffusion, local excursion*, static, oscillations | Regular positioning ‡ | Regular positioning |

*The Brownian ratchet model has all the necessary components/mechanisms to produce regular positioning. However, the parameters used in the study were such that the length-scale was not sufficiently high for regular positioning and therefore only 'local excursion' (approximate positioning in our terminology) instead of regular positioning was observed in cells with one plasmid.

†In this model of the PomXYZ of *Myxococcus xanthus*, PomZ is the analog to ParA and the PomXY cluster is the analog of the partition complex.

‡There are two stochastic models (with and without ParA filaments) presented in **Ietswaart et al., 2014**. The model including filaments is an extension of the other. Both models are capable of producing regular positioning.

positioning was not observed in either of the two existing molecular-level models of plasmid positioning (Brownian Ratchet **Hu et al., 2017** and DNA relay **Surovtsev et al., 2016a**). Our results therefore indicate that neither model is consistent with the dynamics of F plasmid (and indeed of pB171, see below).

Our model lies between the DNA relay and Brownian Ratchet models in terms of the model ingredients but encompasses, in terms of qualitative outputs, the previous stochastic models as specific cases according to $\lambda$, the ratio of the length scale of ParA-ATP dimer diffusion on the nucleoid and the nucleoid length (**Figure 8A**). At one extreme the DNA relay model (**Surovtsev et al., 2016a**) does not incorporate ParA-ATP diffusion ($\lambda = 0$) and therefore oscillations are the only non-trivial behaviour. On the other extreme, the model of **Ietswaart et al., 2014** includes it but without basal ATP hydrolysis ($\lambda = \infty$) such that only regular positioning is obtained. A model of the PomXYZ system of *Myxococcus xanthus* (**Schumacher et al., 2017**), which positions a protein rather than plasmid cargo, also lies here. The Brownian Ratchet model (**Hu et al., 2017**) on the other hand incorporates dimer diffusion but with a length scale (giving $\lambda = 0.5$) that places the system just outside of the regular positioning regime for the case of a single plasmid. This explains why this study found single plasmids to exhibit 'local excursions' around mid-cell, that is approximate rather than regular positioning (**Figure 3F**). Further comparison of the different stochastic models is given in the Materials and methods section and in **Table 3**.

A recent deterministic model deviates from this scheme (**Jindal and Emberly, 2019**) and requires some explanation. This model does not include ParA diffusion on the nucleoid (hence $\lambda = 0$) but it nonetheless produces regular positioning. This is in contrast to our model, for which we otherwise could not obtain, with biologically relevant parameters, regular positioning (**Figure 3—figure supplement 4**), as well as the DNA relay model. We believe this disparity is due to the continuous nature of the Jindal and Emberly model. The plasmid velocity is determined by the entire ParA dimer concentration but weighted according to the distance from the plasmid. Therefore, there is a regime (in which the plasmid movement is slower than ParA un-/binding) for which the mid-cell position is the stable configuration since at that location the weighted sum of ParA dimers on either side balances. This holds even if the ParA gradient is locally symmetric around an off-centre plasmid and because

the model is continuous, even small differences produce an effect. In contrast, in the deterministic models with ParA-ATP diffusion in the nucleoid, the geometry of the cell is encoded in the local ParA gradient around the plasmid (when $\lambda$ is sufficiently large) and plasmid positioning does not rely on interactions with distant ParA. In our stochastic model without such diffusion ($D_h = 0$), the relatively low concentration of ParA dimers means that the effect of rare long-distance interactions between the plasmid and ParA dimers is not sufficient to affect the dynamics due to the inherent stochasticity of the system. The Jindal and Emberly model also predicts that oscillations occur for intermediate plasmid concentrations i.e. the system transitions from regular positioning to oscillations to regular positioning with increasing plasmid concentration. However, we only observe the latter transition experimentally.

We also note that a previous deterministic model (*Walter et al., 2017*) implemented an alternative scheme in which the finite diffusive length-scale is of cytosolic ParA rather than the nucleoid-associated dimers. From a mathematical viewpoint, this system has very similar dynamics and is capable of both oscillations and regular positioning. However, given the rapid diffusion of small cytosolic proteins and the slow transition of ParA to its DNA-binding competent state (*Vecchiarelli et al., 2010*), we expect that ParA in the cytosol is well-mixed.

In *Figure 8A*, we indicate the location of F plasmid in the one-plasmid phase diagram of our model based on the fit to the subpopulation of cells containing a single plasmid (*Figure 4*). While on average it lies in the regular positioning regime, we have seen that in the longest cells it exhibits confined oscillations. This was predicted by our model since longer cells have lower $\lambda$. We also examined the dynamics of the type 1b ParABS system of pB171 and found clear oscillatory dynamics in the majority of cells carrying a single plasmid. While we do not have enough data for an accurate fitting, the 10–15 min period of the oscillations places pB171 within the indicated region, suggesting that both $\varepsilon$ and $\lambda$ are lower than for F plasmid. While adjusting the model parameters can change the nature (period etc) of the dynamics within the regions, we have found that their locations within the phase diagram are relatively robust (see e.g *Figure 3—figure supplements 3 and 4*). Thus, while additional system parameters may be involved, we speculate that that $\lambda$, and hence the diffusive length scale of ParA dimers, is lower for pB171 than for F plasmid. It remains to be seen if this is the case. Nevertheless, our results indicate that both F plasmid and especially pB171 lie close to the dynamical transition between regular positioning and oscillations, with the systems only crossing into the oscillatory regime for the subpopulation of cells with the lowest plasmid concentration. This was previously suggested but not experimentally demonstrated for F plasmid (*Walter et al., 2017*). Importantly, both systems can be explained by the same model.

To understand why this might be the case, we used our model to measure the consumption of ATP throughout the explored phase space. Interestingly, we found that the oscillatory regime consumed the least ATP (*Figure 8B*). This is because this regime has the least flux of ParA-ATP dimers into the plasmid due to the short distance ParA dimers diffuse before dissociating from the DNA despite the fact that the plasmid moves back and forth across the nucleoid i.e. the directed movement of the plasmids cannot compensate for the reduced incoming flux of ParA dimers. Consistent with this, the oscillatory regime has the fewest simultaneous ParA-plasmid tethers (*Figure 3—figure supplement 2*). That the dynamics due to ParABS lie just below the onset of oscillation may therefore be due to achieving regular positioning while at the same time minimising energy consumption.

Overall, our results uncover the dynamical nature of ParABS systems and propose a unified stochastic model that accurately explains the observed plasmid dynamics and dynamical transitions. This model and the insights gained from it will further our understanding of chromosomal ParABS systems, which share many similarities with their plasmid-based relatives (especially those of type 1 a). In particular, having a clear picture of partition complex dynamics will be useful to untangle the unknown role of CTP in partition complex positioning and segregation.

## Materials and methods
### Strains and growth condition

F plasmid experiments use strain DLT3125 (*Sanchez et al., 2015*), a derivative of the *E. coli* K-12 strain DLT1215 (*Bouet et al., 2005*) containing the mini-F plasmid derivative pJYB234. This plasmid carries a functional ParB-mVenus fusion. Overnight cultures were grown at 37 °C in LB-Media containing 10 µg/ml thymine + 10 µg/ml chloramphenicol.

Experiments on plasmid pB171, use strain SR1 (*Ringgaard et al., 2009*), a derivative of the *E. coli* K-12 strain containing a *ΔpcnB* mutation which reduces the copy number of the hosted pB171-derived plasmids. SR1 carries plasmids pSR233 and pSR124 (*Ringgaard et al., 2009*). Plasmid pSR233 is a miniR1 plasmid carrying the parABS system (*par2*) of pB171 in addition to a *tetO* array. Plasmid pSR124 encodes an inducible *tetR-mCherry* fusion under the control of a $P_{BAD}$ promoter. TetR binds to *tetO* and allows to track the motion of pSR233. Overnight cultures were grown at 37 °C in LB-Media containing 1 µg/ml thiamine + 50 µg/ml kanamycin + 100 µg/ml ampicillin.

## Microfluidics

Like the original mother machine (*Wang et al., 2010*), our design consists of a main channel through which nutrient media flows and narrow growth-channels in which cells are trapped. However, we follow (*Baltekin et al., 2017*) and include (i) a small opening at the end of each growth channel (ii) a waste channel connected to that opening to allow a continuous flow of nutrients through the growth channels (iii) an inverted growth-channel that is used to remove the background from fluorescence and phase contrast. We used a silicon wafer with this design to create the mother machine. We poured a polydimethylsiloxane (PDMS) mixture composed of a ratio of 1:7 (curing agent:base) over the wafer and let it rest at low pressure in a degasser for ~30 min to remove air bubbles inside. The PDMS was then baked at 80 °C overnight (~16 h). The cured PDMS was peeled off the wafer. Before imaging, the chip is bonded to a glass slide using a plasma generator (30 s at 75 W) and subsequently baked for a further 30 min at 80 °C, while the microscope is prepared.

## Microscopy

We used a Nikon Ti microscope with a 100 x/1.45 oil objective and a Hamamatsu Photonics camera for all imaging. For imaging cells of strain DLT3125 we used a mother machine. Overnight cultures were inoculated into fresh media (M9+0.5% glycerol + 0.2% casamino acids + 0.04 mg/mL thymine + 0.2 mg/mL leucine + 10 µg/mL chloramphenicol) for 4 hours at 30 °C before imaging. Cells were loaded into the chip through the main channel and the chip was placed into a preheated microscope at 30 °C. The cells were constantly supplied with fresh media by pumping 2 µL/min of M9+0.5% glycerol + 0.2% casamino + 0.04 mg/mL thymine + 0.2 mg/mL leucine through the microfluidic chip. Cells were grown for 2 hr inside the microscope before imaging. Cells were imaged at 1 minute intervals for approximately 72 hr. Both phase contrast and YFP-signal were captured. Imaging was repeated independently with similar results.

For imaging cells of strain SR1 we used agar pads. Overnight cultures were inoculated into fresh media (M9+0.5% glycerol + 0.2% casamino + 1 µg/ml thiamine + 10 µg/ml arabinose + 50 µg/ml kanamycin + 100 µg/ml ampicillin) for 2 hr at 30 °C before imaging. The arabinose was added to induce synthesis of *tetR-mCherry*. Longer or continuous induction of arabinose leads to replication defects. Cells were placed on an 1% agar pad made from M9+0.5% glycerol + 0.2% casamino acids + 1 µg/mL thiamine and they were imaged at 1 min intervals for 4 hr. Both phase contrast and RFP-signal were captured. Imaging was performed twice and the data combined.

## Image processing

Our image processing pipeline for mother-machine experiments consists of three parts: (I) preprocessing, (II) segmentation and foci finding, and (III) cell and foci tracking. While Parts I and III use custom Matlab scripts, Part II is based on SuperSegger (*Stylianidou et al., 2016*), a Matlab-based package for segmenting and tracking bacteria within microcolonies (original code is available at https://github.com/wiggins-lab/SuperSegger; *Wiggins, 2018*), that we modified to better handle high-throughput data. SuperSegger employs pre-trained neural networks to segment cells by identifying their boundaries. It comes with a pre-trained model for *E. coli* which worked very well with our data. Therefore there was no need to train our own neural network. SuperSegger is capable of tracking cells however the tracking did not work properly with mother-machine images and so we developed our own method. Nevertheless, acknowledging that one of the main components of our pipeline, the segmentation, uses SuperSegger we refer to the entire pipeline as MotherSegger (code is available at https://gitlab.gwdg.de/murray-group/MotherSegger/-/tree/PaperParABS; *Koehler and Murray, 2022b*; copy archived at swh:1:rev:42e2a6ec49b12fc19fe14e3fe2247f699f110f9e).

In Part I, each frame of an acquired image stack is aligned (the offset between frames in x and y is removed). Afterwards the image stack is rotated so the growth channels are vertical. A mask of the mother machine layout is fitted to the phase contrast, using cross-correlation, to identify where the growth channels are located. Each growth channel is extracted from the image stack and the flipped inverted channel is subtracted to remove the background from both the fluorescence signal and phase contrast. The images are then segmented and fluorescent foci are identified using Supersegger.

In Part III, both foci and cells are tracked. Since cells cannot change their order inside the growth channel, they can be tracked by matching similar cell length between frames (starting from the bottom of the growth channels). Once individual cell cycles are identified, the foci positions found by Supersegger are re-specified relative to the bounding box of the cell (the smallest rectangular image containing the cell mask) on each frame. Since cells are vertical in the channels without any significant tilting, the bounding box is aligned with the cell axes. Within each cell cycle, foci are tracked between frames by finding the closest focus on the next frame inside the same cell cycle. The effect of growth on foci position was neglectable since cells grew on average much less than one pixel per frame at the 1 min frame rate and 100 min doubling time used here. Finally, half the cell length was subtracted from the foci positions along the long cell axis (vertical direction) so that 0 corresponds to the middle of the cell. The sign of the positions was also adjusted so that negative positions refer to the old-pole proximal side of the cell.

To filter out potential segmentation errors, cell cycles that do not have exactly 1 parent and 2 daughters are excluded from analysis along with their immediate relatives (with the exception of those who are pushed out of the growth channel). For the analysis of foci trajectories, we considered only trajectories coming from at least 12 consecutive frames with the same number of foci. For pB171, we used (unmodified) SuperSegger to process images of cells growing on agarose pads.

## Over-damped spring

The distribution $p\left(x, \delta t | x_0\right)$ describes the probability that a Brownian particle, initially at position $x_0$, experiencing a spring-like force (harmonic potential) towards 0 is found at position $x$ at a time $\delta t$ later (***Doi and Edwards, 1988***):

$$p\left(x, \delta t | x_0\right) = \sqrt{\frac{f/k_B T}{2\pi S}} exp\left[-\frac{f/k_B T}{2S}\left(x - x_0 e^{-\delta t/\tau}\right)^2\right] \text{ where } \boldsymbol{S} = 1 - \boldsymbol{e^{-2\delta t/\tau}} \text{ , } \tau = \frac{k_B T}{fD} \text{ , } k_B \text{ is Boltzmann's}$$

constant and T is the absolute temperature. The stiffness of the spring is $f/k_B T$ and D is the intrinsic diffusion coefficient. From this, it is straightforward to calculate the expected value and variance of the step-wise velocity $v := \frac{x - x_0}{\delta t}$ to be $E\left[v\right] = \frac{e^{-\delta t/\tau} - 1}{\delta t} x_0$ and $Var\left[v\right] = \frac{D\tau}{\delta t^2}\left(1 - e^{-2\delta t/\tau}\right)$. Note the two properties characteristic of an (over-damped) spring-like force: The expected value $E\left[v\right]$ linearly scales with $x_0$ while the variance $Var\left[v\right]$ is independent of the initial position. We observed the same properties in our experimental data. We determined D and $\tau$ via $D = \frac{Var[v] Ln(\delta tm + 1)}{\delta tm^2 + 2m}$ , $\tau = -\frac{\delta t}{ln(\delta tm + 1)}$ where m is the slope of the velocity profile.

The distribution $p\left(x, \delta t | x_0\right)$ can also be used to calculate the position and velocity autocorrelations:

$$E\left[x\left(t_0\right) x\left(t_0 + t\right)\right] / E\left[x\left(t_0\right)^2\right] = e^{-t/\tau} \quad \text{and} \quad E\left[v\left(t_0\right) v\left(t_0 + t\right)\right] / E\left[v\left(t_0\right)^2\right] = \frac{2e^{-t/\tau} - e^{-|t-\delta t|/\tau} - e^{-(t+\delta t)/\tau}}{2 - 2e^{-\delta t/\tau}}$$

respectively. Finally, a characteristic force can be defined as the force on the particle at an extension of one standard deviation of the equilibrium distribution i.e. at $x = \sqrt{k_B T/k}$ . For F plasmid, this gives a force of $F = k\sqrt{k_B T/k} = 0.019\ pN$ at $T = 30°C$.

## Model

Our model is an extension of the previous DNA-relay model (***Surovtsev et al., 2016a***) that incorporates diffusion on the nucleoid (hopping) and basal hydrolysis of ParA-ATP and uses analytic expressions for the fluctuations rather than a second order approximation. Like the DNA relay it is a 2D off-lattice stochastic model and updates positions in discrete time steps $dt$. The implementation was written in C++ (code is available at https://gitlab.gwdg.de/murray-group/hopping_and_relay/-/tree/PaperParABS; ***Koehler and Murray, 2022a***; copy archived at swh:1:rev:75dc93a972652847a2ea-30bada3bc3206568edfa). It consists of the following components. ParA associates to the DNA non-specifically in its ATP-dependent dimer state with the rate $k_a$ . Once associated, ParA (i.e. ParA-ATP dimers) moves in two distinct ways: (i) Diffusive motion on the nucleoid with the diffusion coefficient $D_h$ . This is an effective description of the movement of dimers due to transient unbinding events that allow them to 'hop' between DNA-strands. We do not consider the alternative scenario in which

dimers transfer between DNA strands when the latter come into contact. In this scenario the effective diffusion coefficient would depend on the parameters describing the DNA fluctuations ($D_A$ and $\sigma_{x,y}$). (ii) Between hopping events, each bound ParA dimer experiences the elastic fluctuations of the DNA strand it is bound to. This is implemented as elastic (spring-like) fluctuations around its initial position. Dimers dissociate from the nucleoid due to either basal ATP hydrolyse at a rate $k_d$ or due to hydrolysis stimulated by ParB on the plasmid. The latter is modelled as a ParB-coated disc and ParB-ParA tethers form whenever the disk comes in contact with a ParA dimer. ParB-stimulated hydrolysis then breaks these tethers at a rate $k_h$ , returning ParA to the cytosolic pool. The plasmid experiences the elastic force of every tethered ParA and moves according the its intrinsic diffusion coefficient $D_p$ and the resultant force of all tethers. An overview of this scheme is shown in *Figure 3A*.

As in the DNA relay model we have made some simplifications that we next make explicit. First, we only modelled three states of ParA: 'nucleoid associated' and 'cytosolic' and 'tethered'. Second, cytosolic ParA are assumed to be well mixed. This is justified based on the slow conformation changes needed to return it to a state competent for DNA-binding (*Vecchiarelli et al., 2010*). Third, no individual ParB molecules were modelled, rather the plasmid is treated as a disk coated with enough ParB that each nucleoid bound ParA that makes contact with the plasmid instantaneously finds a ParB partner, therefore removing the need to model individual ParB. This is justified by the substantially higher local concentration of ParB compared to ParA at the plasmid.

The nucleoid is modelled as a rectangle with the dimensions $L \times W$. The positions of ParA and the plasmid(s), are updated every time step $dt$ as follows. Between hopping events, each nucleoid associated ParA dimer fluctuates about a home position $x_h$ . The new position $x(t + dt)$ of each dimer is given by $x(t + dt) = x_h + \delta x$, where $\delta x$ is drawn with probability $p(\delta x, dt \mid x(t) - x_h)$ where $x$(t) is its original position (see section 'Over-damped spring') and the normalised spring constant ($f/k_B T$ above) along each dimension is $1/\sigma_{x,y}^2$ and the diffusion coefficient $D_A$ . During hopping events $x(t)$ and $x_h$ are both offset by a value drawn from a Gaussian distribution with $\mu = 0$ and $\sigma = \sqrt{2 D_h dt}$ for both dimensions. The displacement of the plasmid is determined similar to each ParA dimer but according to the resultant force acting on it. This resultant force vector has an effective spring constant equal to the spring constant of a single tether times the number of tethers and acts towards an equilibrium position $x_p(t) + \sum_{tethers} (x_h - x(t)) / n$, where $x_p(t)$ is the plasmid position and the sum is over all ($n$) tethers. We ignore the effects of Torque. The intrinsic diffusion coefficient of the plasmid is $D_p$ . If the plasmid has no tethers attached then it moves by normal diffusion, with displacements drawn from a Gaussian distribution with $\mu = 0$ and $\sigma = \sqrt{2 D_p dt}$ . The x and y components of all positions are updated independently and all simulations in this paper were run until the system reached equilibrium before acquiring data used for analysis.

## Comparison of stochastic ParABS models

The most recent stochastic models of positioning by ParABS and ParABS-like systems explicitly incorporate earlier proposals for the mechanism of force generation, namely, that the elastic fluctuations of the DNA and/or ParA-ParB protein tethers can power the movement of cargo up the gradient of DNA-bound ParA-ATP dimers (*Hu et al., 2015*; *Lim et al., 2014*). However, the models differ in other ways (see *Table 3*).

The DNA relay model (*Surovtsev et al., 2016a*) does not allow DNA-bound ParA dimers to diffuse (hop) on the nucleoid. They fluctuate around a home position due to the elastic fluctuations of the underlying chromosomal locus. In our terminology, this model therefore has $\lambda = 0$, where $\lambda$ is the ratio of the ParA dimer diffusive length scale to the nucleoid length (see main text). Oscillations were the only non-trivial behaviour found in this model.

The Brownian Ratchet model (*Hu et al., 2017*) on the other hand includes diffusion of dimers on the nucleoid as well as several other details such as explicit modelling of the transient DNA-bound ParA-ADP state, limited binding to cargo and the force and length-dependent dissociation of ParA-ParB tethers. It also includes basal (plasmid independent) ParA hydrolysis. Together with diffusion on the nucleoid, this results in $\lambda$ being finite. However, through parameters analyses, its value was fixed at $\lambda = 0.5$. As a result, 'local excursions' (confined diffusion in our terminology) were observed for single plasmids rather than regular positioning. This led the authors to conclude that the biological system lies in a narrow regime of the model parameter space in which two or more plasmids are regularly positioned due to newly replicated plasmids moving apart a fixed distance ('directed segregation').

While not incorporating an explicit mechanism of force generation, the earlier model of *Ietswaart et al., 2014* is worth mentioning here. This stochastic model was based on the formation of short DNA bound ParA-ATP filaments. It included diffusion of ParA dimers on the nucleoid but without basal hydrolysis ($k_d = 0$). Hence, the diffusion of each ParA dimer on the nucleoid is interrupted only upon interaction with a plasmid and $\lambda = \infty$. This model gave regular positioning, as did a variant without ParA filament formation, as the only non-trivial behaviour. The authors explained the emergence of regular positioning by a 'flux balance' argument. Plasmids are positioned because that is the unique configuration in which the diffusive flux of ParA dimers into each plasmid from either side balances (see *Figure 2* and main text). They demonstrated this mathematically using a simplified deterministic model in which plasmids both act as sinks for ParA-ATP and move up the gradient of ParA-ATP on the nucleoid.

The above models have also been applied to ParA-like systems. In the PomXYZ system of *Myxococcus xanthus*, the ParA-like protein PomZ positions a large protein cluster formed by PomX and PomY at the middle of the cell. (*Schumacher et al., 2017*) explained this behaviour using the same elastic DNA/protein bond fluctuations as the models above, combined with the flux-balance mechanism of Ietswaart et al. Like the latter, their model did not include basal hydrolysis of PomZ and therefore $\lambda = \infty$. More recently, the Brownian Ratchet model has also been used to explain the positioning of carboxysomes in the cyanobacterium *Synechococcus elongatus* (*MacCready et al., 2018*).

## Phase space

To generate the phase space (*Figures 3I and 8A*, *Figure 3—figure supplement 1A*) of our model we chose 100 values of $\lambda$ and $\varepsilon$, resulting in a 100 by 100 grid of different parameter combinations. This was done by varying $D_h$ for $\lambda$ and $k_h$ for $\varepsilon$. To visualise the behaviour of each parameter combination we considered three quantities (i) $\varphi$, (ii) $\psi$ and (ii) $\chi$. (i) φ describes the goodness of regular positioning. The mean position of a plasmid-trajectory along the long axis is used as an input to a triangle wave function: $\varphi = f_n\left(x\right) = 1 - \left|2\left(nx/L - \lfloor nx/L \rfloor - 0.5\right)\right|$, where $x$ is the mean position of the trajectory, $L$ is the cell length and $n$ is the number of plasmids. If the mean of a trajectory is equal to the position defined by regular positioning for that number of plasmids, then $\varphi$=1. If a trajectory is positioned at a pole or exactly between two regular positions it returns 0. (ii) $\psi$ describes the mobility of a trajectory. $\psi$ is equal to the standard deviation of the trajectory positions divided by $L/\left(\sqrt{12}n\right)$ (the standard deviation of the uniform distribution of width L/n). If a trajectory is oscillating, its distribution of position is roughly uniform (*Figure 3E*) resulting in a $\psi$ close to 1. If a trajectory stays approximately at one position, $\psi$ is much lower than 1. (iii) $\chi$ describes if a trajectory is oscillating. $\chi$ is equal to the highest positive maxima after the first negative minima in the normalised autocorrelation function of position. $\chi$ is equal to 0 if there is no negative minima or no positive maxima after a negative minima. From these three quantities, we calculated three descriptors with values between 0 and 1, describing the three regimes of oscillations, regular positioning and static. Regular Positioning: $\left(1 - \psi\right)\varphi$ is high if the plasmid is non-mobile and regularly positioned. Static: $\left(1 - \psi\right)\left(1 - \varphi\right)$ is high if non-mobile and not regularly positioned. Oscillations: $\chi$ is high if oscillating. Each regime is associated with a colour and this colour is scaled by its corresponding descriptor. The colours were chosen to be colourblind friendly (light brown RGB:[255 193 7], blue RGB:[30 136 229], pink RGB:[216 27 96]). Note that for diffusive trajectories, we expect $\varphi = 1$, $\psi = 1$ and $\chi = 0$ and hence all descriptors are 0 (visualised as black). To smoothen the phase diagram we used the morphological operation 'opening' followed by a 2-D Gaussian filter. The methodology above, while somewhat arbitrary, was found to describe the dynamics of the system well.

## Classification of trajectories

Regular positioning and oscillations are distinguishable by calculating the velocity autocorrelation between adjacent frames (lag 1 min, *Figure 1—figure supplement 2A*). However, this method does not work for trajectories which change behaviour. Therefore, we developed a procedure to find segments inside a trajectory which are oscillatory or regularly positioned. A sliding window is moved across a trajectory and the velocity autocorrelation is calculated inside the window. If the autocorrelation at lag 1 is positive the point in the middle of the sliding window is annotated 'oscillatory'. Otherwise the point is labelled 'regularly positioned' (*Figure 6—figure supplement 1A, B*). Multiple points in a row with the same annotation form a segment (*Figure 6—figure supplement 1B*). With this

procedure a trajectory can be broken down into multiple segments belonging to different behaviours (*Figure 6—figure supplement 1C*). To annotate our data we used a sliding window of size 12 and a requirement of 6 successive points of the same annotation to form a segment (*Figure 6—figure supplement 1D*). One weakness of this approach is the small window size which may result in false positives oscillatory segments.

## Acknowledgements

We thank Jean-Yves Bouet (Toulouse), Christine Jacobs-Wagner (Stanford), Yong Zhang (Copenhagen), Kenn Gerdes (Copenhagen) for providing strains and plasmids.

## Additional information

### Funding

| Funder | Grant reference number | Author |
|---|---|---|
| Max-Planck-Institut für Terrestrische Mikrobiologie | Core funding | Seán M Murray |
| Deutsche Forschungsgemeinschaft | MU 4469/2-1 | Robin Köhler Seán M Murray |

The funders had no role in study design, data collection and interpretation, or the decision to submit the work for publication.

### Author contributions
Robin Köhler, Data curation, Software, Formal analysis, Investigation, Visualization, Methodology, Writing - original draft; Eugen Kaganovitch, Methodology; Seán M Murray, Conceptualization, Data curation, Formal analysis, Supervision, Funding acquisition, Investigation, Methodology, Writing - original draft, Project administration, Writing - review and editing

### Author ORCIDs
Seán M Murray ⬦ http://orcid.org/0000-0002-2260-0774

### Decision letter and Author response
Decision letter https://doi.org/10.7554/eLife.78743.sa1
Author response https://doi.org/10.7554/eLife.78743.sa2

## Additional files

### Supplementary files
• MDAR checklist

### Data availability
Plasmid tracking data has been deposited to Edmond, the Open Research Data Repository of the Max Planck Society and are available at https://doi.org/10.17617/3.UKEOIU. Code is available at the gitlab repositories indicated in the Materials and Methods.

The following dataset was generated:

| Author(s) | Year | Dataset title | Dataset URL | Database and Identifier |
|---|---|---|---|---|
| Köhler R, Murray S | 2022 | Plasmid tracking | https://doi.org/10.17617/3.UKEOIU | Edmond, 10.17617/3.UKEOIU |

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
