## [Editor Report]

This study provides new experimental data and detailed modeling of the partitioning of low copy plasmids under the control of the ParABS system in bacteria. The dynamics of the partition complex is tracked over many generations, providing valuable data to constrain the models. The authors propose a compelling model which can manifest either regular positioning or oscillations depending on the model parameters. The research will be of interest to biologists and biophysicists interested in cellular dynamics and internal organization in bacteria.

---

## [Decision Letter]

**Decision letter after peer review:**

Thank you for submitting your article "High-throughput imaging and quantitative analysis uncovers the nature of plasmid positioning by ParABS" for consideration by *eLife*. Your article has been reviewed by 2 peer reviewers, and the evaluation has been overseen by a Reviewing Editor and Aleksandra Walczak as the Senior Editor. The reviewers have opted to remain anonymous.

Essential revisions:

Both reviewers believe that the paper has the potential to be published in *eLife*, but have substantive comments that should be addressed, as detailed below and in their reviews. Please note that both reviewers did not hinge the publication on additional experiments.

(1) Not all claims are fully supported by the presented data, in particular the claim of the role of ParA hopping/diffusion.

(2) Limited analysis of the control parameters.

(3) Unsatisfactory analysis on the origin of the difference between F1 and pB147 plasmids dynamics.

(4) More careful comparison/analysis to previously published model of ParA and ParA-like systems is an essential element needed to make this work impactful.

(5) (Optional) Providing information on the ParA distribution would be a very strong addition.

*Reviewer #1 (Recommendations for the authors):*

Using "hopping" as a substitute for the ParA diffusion over the chromosome and then stating that it was "primary determinant of the geometry sensing" (abstract) might be misleading. What the authors did – they considered ParA diffusion in the model. And that apparent diffusion over the chromosome might be result of at least two non-exclusive scenarios – repeated cycles of binding/unbinding of ParA dimer intermittent by the diffusion of the unbound ParA dimer or direct hopping of chromosome-bound ParA from one chromosome locus to another when they come into contact upon their intrinsic fluctuations.

Along the same line, stating in the Abstract that "we identify ParA hopping on the nucleoid as the primary determinant of this geometry-sensing" is not correct as neither the hopping was explicitly considered in the model nor the author really tested whether this statement would be correct even for ParA diffusion. The only test involved was analysis of the motion in the short cells vs long cells, without perturbation of diffusion per se. Moreover, the authors observed that in case of pB171 plasmid mode of motion was different, yet it is not clear whether the difference could be explained by the model as due to a difference in diffusion coefficient or kd or something else. This reviewer believes that "identify" requires a little bit more that being able to reproduce observed behavior by changing a parameter in the model.

"Lack of quantitative experiments" mentioned several times by the authors might not be exactly the case. While previous experiments/analysis was not the same what the authors did, several groups measured different experimental metrics (just a few examples, far from exhaustive, – Li et al. 2004 Mol.Microbiol, Surovtsev et al. PNAS 2016, Le Gall et al. Nat.Commun. 2016)

The authors compare their model to "diffusive" and "superdiffusive" models (Figure 1 Figure Sup 2A), but details on how they were modeled are lacking.

p.5 ln 1-3 The authors report characteristic timescale, τ, at which elastic fluctuations act to be about 120 s using fitting of the velocity vs position data, and then they report that the same τ is ~ 170 s using velocity and position autocorrelation functions fitting, concluding the values are comparable. That seems to this reviewer as quite a difference, warranting at least some comment on the potential origin of the difference.

Regarding velocity autocorrelation function and positional dependence, it would be really helpful and reassuring to calculate them from the higher temporal resolution (dt=1s mentioned for MSD and D calculations) since the authors already have the data.

p.5 ln 33 …have previously shown that regular positioning can theoretically be achieved, independently of the particular mechanism of force generation, through the balancing of the diffusive fluxes. Given that the force is what really defines where the cargo moves, I don't think the positioning mechanism can be dissected from the mechanism of force generation, once one tries to conclude what specific mechanism operates for a given experimental system. For example, in the model the authors simulated here the force is not directly dependent on the flux of the ParA on the plasmid, rather it depends on the local distribution of ParA. While it is not necessarily negates the authors reasonings, it does require an additional explanation on how these reasonings relates to the simulated model…

p.6 Figure 2 (A) When *s* ≪ *L*/2 (i), where *L* is the nucleoid length, a disparity in the flux only exists very close to the poles (blue region). This seems somewhat counterintuitive, as this regions actually many s away from the sink of ParA (i.e. plasmid)…

p.7 30-33 Imaging studies in several Par systems, especially those that position non-DNA cargos, have observed that ParA fluorescence can be higher at the plasmid than elsewhere (Roberts et al., 2012; Schumacher et al., 2017). This is in somewhat disagreement with the canonical picture of the ParB coated cargo acting as a sink for ParA-ATP. This is not a real conundrum, as previous models showed this effect (Surovtsev et al. Biophys.J. 2016, Hu et al.Biophys.J. 2021)

Figure 3I What is the color code? It is actually described deep in the methods, but it would be really useful to have it in the main text or figure legend.

Figure 3 Sup.Figure -2B what is the color code?

In image analysis description, the authors do not provide any details beyond referring to the general Segger description and MotherSegger code on the most important part – cell segmentation and defining position of the plasmid. This reviewer believes that some short description should be readily available within the text for the reader to understand potential limitations. For example, beyond just finding position in the image, how it was used for the analysis – was it position in image coordinate or relative to the cell coordinate, and how change in the coordinate, without motion due to cell growth was taken into account.

It seems that the number of ParA and spring constant values are not specified for the model.

*Reviewer #2 (Recommendations for the authors):*

The paper claims that they are the only paper to have a model that shows regular positioning of the ParABS system and that models without substrate hopping on the nucleoid only admit oscillations. This is not true. Jindal and Emberly (2019) showed that regular positioning of plasmids could occur in a model that did not allow for any diffusion of substrate in the nucleoid and that oscillations would emerge due to relaxing of confinement or potentially the liberation of substrate resources due to the addition of plasmids. Indeed the phenomena observed in these experiments (regular positioning, transitioning to oscillations, and back to regular positioning) was predicted in that paper. Have the authors fully explored the parameter space of their model? If they set kh = 0, (i.e. no hopping), are there any values of n_A, and on/off rates that allow for regular positioning that transitions to oscillations as the cell lengthens? For regular positioning, it requires a broad wake that is balanced between left and right. On longer cells, the confinement is relieved and the complex can oscillate. It would be interesting to know if the stochastic formulation of the model does not allow for any regular positioning if kh=0. If it does, are the parameters values such that they are completely inconsistent with measured kinetic parameters, thus necessitating hopping for the given system.

A few other comments/questions:

I'm assuming Figure 6 is from experimental data, but there are no reported cell numbers for the various distributions and statistics.

it would have been nice to have seen data from > 2 plasmids. Do the authors ever see oscillations in 2 plasmids switching to regular positioning once 3 plasmids are present (i.e. Figure 7F with a column for 3 plasmids). Presumably yes as there are around ~20% of the 2 plasmid systems oscillating, and when 3 are present, regular positioning likely follows. Do they ever get filamentous cells, and what are the dynamics like in those cells?

I am intrigued by the difference in dynamics for the F-plasmid and pB171 plasmid. Their experimental results for the 1b system show it is more likely to oscillate. Why? Is it due to a smaller s? The paper claims that it is due to smaller s, but no real discussion/evidence is given.

I could find no details of how varying n_A affects results. As in most other published models, this also has a huge effect on dynamics, similar to their parameter, λ. Could some of their observations be due to cell-to-cell heterogeneities in n_A? Also dilution would have an effect, which it is not clear if it is taken into account here. Do they use n_A=500 for all simulated cell lengths? Could differences in the total amount of ParA explain the different dynamics between the F plasmid and pB171 (see my comment above)?

Have they done lineage tracking? Do they see correlations in the likelihood to do regular positioning or oscillations? If so, especially for the case with 1-plasmid oscillations, is it due to length differences in the daughter? or could oscillations be arising from some other unmeasured system parameter?

[Editors’ note: further revisions were suggested prior to acceptance, as described below.]

Thank you for resubmitting your work entitled "High-throughput imaging and quantitative analysis uncovers the nature of plasmid positioning by ParABS" for further consideration by *eLife*. Your revised article has been evaluated by Aleksandra Walczak (Senior Editor) and a Reviewing Editor. Please take note of the points below and address them in the final version submitted.

*Reviewer #2 (Recommendations for the authors):*

Summary:

This work focuses on how subcellular objects (plasmids) can sense spatial dimensions of the cell and how they are transported to the specific targeted positions. The authors expand previously proposed 'DNA-relay' mechanism of the intracellular transport in which plasmid are actively driven by chromosome fluctuations. By adding diffusion of chromosome-bound ParA, a protein that 'links' the plasmid to fluctuating chromosome, to the model, they show that this expanded model can reproduce full range of observed experimental dynamics including plasmids oscillations and direct motion to the mid-cell. This work reconciles some apparent differences between previously published models on ParA-dependent intracellular transport and expand our understanding of the chromatin-fluctuation-driven intracellular patterning.

Review:

Kohler and Murray present high-throughput image-based measurements of how low-copy F plasmids move (segregate) inside *E. coli* cell. This active segregation ensures that each daughter cell inherit equal share of the plasmids. Previous work by different labs have shown that faithful F-plasmid segregation (as well as segregation of many other low-copy plasmids, segregation of chromosomes in many bacterial species and segregation of come supramolecular complexes) requires ParA and ParB proteins (or proteins similar to them) and is achieved by an active transport mechanism. ParB is known to bind to the cargo (plasmid) and ParA forms a dimer upon ATP binding that binds to DNA (chromosome) non-specifically, and also can bind to ParB (associated with cargo). After ATP hydrolysis (stimulated by the interaction with ParB), ParA dimer dissociates to monomers and from ParB and the chromosome. While different mechanisms of the ParA-dependent active transport had been proposed, recently two mechanisms become most popular – one based on the elastic dynamics of the chromatin (Lim et al. *eLife* 2014, Surovtsev PNAS 2016, Hu et al. Biophys.J 2017, Schumaher Dev.Cell 2017) and the other based on a theoretically-derived "chemophoretic" force (Sugawara and Kaneko Biophysics 2011, Walter et al. Phys.Rev.Lett. 2017).

Measuring motion of F plasmid in large number of cells with one or two plasmids allowed authors to overcome inherently stochastic nature of the motion and to analyze plasmid spatial distribution, plasmid displacement (i.e. velocity) as a function of their relative position, and autocorrelations of the position and the displacement. They concluded that these metrics are consistent with 'true positioning' (i.e. average plasmid displacement is biased toward the target position – center for one plasmid and 1/4 and 3/4 positions for two plasmids) but not with 'approximate positioning' (i.e. when plasmid moves around target position, for example, in near-oscillatory fashion). This 'true positioning' can be described as a particle moving on the over-dampened spring. They reproduce this behavior by expanding previous model for 'DNA-relay' mechanism (Lim et al. *eLife* 2014, Surovtsev PNAS 2016), in which plasmid is actively moved by the elastic force from the chromosome and ParA serves to transmit this force from the chromosome to the plasmid. Now, the authors explicitly consider in the model that the chromosome-bound ParA can diffuse and this allows the model to achieve 'true plasmid positioning' for some combination of model parameters in addition to oscillatory dynamics reported in the original model.

Based on their computational model, the authors proposed that two parameters: (1) diffusion scale of ParA, i.e. typical length diffused by ParA before dissociation, λ = 2(2Dh/kd)1/2/L (here, Dh is diffusion rate of DNA-bound ParA, kd is ParB-independent hydrolysis rate, i.e. lifetime of DNA-bound ParA); and (2) ratio of ParB-dependent and -independent hydrolysis rates epsilon = kh/kd; are key control parameters defining what qualitative behavior is observed. By varying these parameters (via changing ~30- and ~200-fold Dh and kh) they showed that their model encompasses all observed dynamic behaviors – random diffusion, near-oscillatory behavior, or overdamped spring ('true positioning'), and illustrated how dynamics of the system changes between these 3 modes of motion. The parameter analysis includes also changing other parameters such as ParA number, elastic spring of the chromosome, etc. for some selected initial combinations of the λ and epsilon.

The authors also show by simulations that overdamped spring dynamics can transition into oscillatory behavior when λ decreases, for example by cell growth. Indeed, they observed more oscillatory behavior when they compared single-plasmid dynamics in the longer cells compared to the shorter cells. This was not the case in double-plasmid cells, in perfect agreement with their analysis. The authors concluded that the system operates close but below (perhaps, "above" should be used as it refers to bigger λ) the threshold to oscillatory regime. The authors also calculated ATP consumption in the model and found that oscillatory regime minimizes ATP consumption.

I think the major impact of the paper is that the expanded model and analysis presented here shows how various dynamics (observed experimentally) can be achieved within the same mechanism in which an intracellular cargo is moved by the fluctuating chromosome via ParA-mediated attachments. While original "bare-bone" DNA-relay model could explain active transport of the plasmid cargo, taking into account diffusion of DNA-bound ParA dimer (and in appropriate value range) was essential to achieve "true positioning" observed for F-plasmids. Importantly, parameters analysis shows how the expanded model encompasses, depending on combinations of control parameters, previously modelled 'oscillations' (Surovtsev PNAS 2016), 'local excursions' (Hu et al. Biophys.J 2017) and 'true positioning' (Schumaher Dev.Cell 2017).

Overall, I think, the revised manuscript unifies previous modelling efforts on ParA/ParB and similar (PomXYZ) systems and clarifies role of ParA diffusion in the dynamic behavior. It advances our general understanding of how out-of-equilibrium dynamics of ParA ATP cycle allows to achieve various modes of intracellular dynamics depending on parameters combination. In a broader perspective, it advances our general knowledge of intracellular organization and of DNA segregation.

Suggestions/Questions:

While the revised manuscript now really helps the reader to understand how ParA/ParB system works, thanks to explicit comparison to earlier models, here are a few things that could be addressed by the authors (in this reviewer opinion).

Again, there is no doubt that λ is an important parameter of the model, however I found the authors explanation (Figure 2) confusing (at least for me). They argue about importance of the parameter based on the importance of the balancing ParA fluxes to the plasmid. But these fluxes would be there and would be balanced only in the center (for one plasmid) no matter how big a nucleoid is relative to the ParA diffusion scale… Moreover, the plasmid interacts with ParA bound mostly within few σ (range of chromosome fluctuations), so argument about "information" also does not work out… Also, the authors did not really test whether λ (but not Dh alone) governs dynamics. The authors varied λ and epsilon independently by changing Dh and kh, but does plasmid dynamics look exactly the same if we say change instead Dh, kd, kh and L such that λ and epsilon do not change? Other parameters sweeps that were added to the manuscript are very appreciated, but they do not answer this question. Along the same line, a bit more expanded discussion on underlying nature of the transition to different dynamics during these sweeps may help reader to understand interplay between different parameters in determining dynamics qualitatively. I found description of results of these sweep too brief (so not very insightful).

Along the same line, it might be a bit counter-intuitive that the system behavior almost does not depend on number of ParA. For example, the authors argued for the importance of λ based on ParA-fluxes, but value of the fluxes should strongly depend on the amount of ParA in the system. Additionally, the authors report that the plasmid velocity strongly depends on the ParA amount associated with it (Figure 3—figure supplement 2 B) (perhaps, on overall amount of ParA as well). One might think that the velocity would play a role whether we observe a strongly dampened spring or a decaying oscillator. Maybe it is a naïve thinking, but, perhaps, this warrants some explanation in the manuscript. And, while for relatively high ParA number the dependence might be saturating, Figure 3—figure supplement 3 top-left and Figure 3—figure supplement 4D suggest that lowering ParA may drive switch to different dynamics.

Additional comment on parameter sweeping. Since ParA diffusion is an "effective" description of some underlaying dynamics, effective Dh might depend on other parameters, i.e. cannot be varied independently. For example, in a potential "binding-unbinding-bulk_diffusion-binding" scenario, Dh depends on k_dis, ka and D_bulk. In an alternative scenario, where ParA hopes (without unbinding) to a new DNA position, Dh depends on σ and Da. While ideally these scenarios should be modelled explicitly to test how changes in these parameters affect apparent Dh and plasmid dynamics, such a limitation, perhaps, should be mentioned in the manuscript (since it will make the model more complex and also there is only so much we can do at once).

Regarding pB171 plasmids, having a plot similar to Figure 1E and F would be nice, as they were used as an evidence for 'true positioning' regime. And still some potential explanation of what might be different between F and pB171 plasmids – D, kd(?) – would be a nice addition as it might prompt someone to test it in the experiment.

On presentation:

I found the title of the paper too vague, as "nature of plasmid positioning" could be interpreted very differently (and thus whether the work "uncovers" it or not).

I would suggest adding to the abstract what were the key ingredients of the authors model to succeed in achieving full range and transition between different modes of the plasmid dynamics.

Still in this reviewer opinion, "lack of quantitative measurements of plasmid dynamics" (used several times through the paper) might be misleading as the authors measured from microscopy exactly the same thing – position vs time – as earlier works. The strength of the work is not in a measuring experimentally a new thing, but having a great statistics (high-throughput imaging!) that enabled a new analysis (meaningful beyond inherent stochastic noise) – velocity vs position and velocity and position autocorrelation functions. I would emphasize this achievement instead.

Figure 1E referred before any other panels, and similar happens with some other panels through the manuscript.

Table1: I do not think Weber et al. 2010, or Javer et al. 2014 reported any chromosomal spring constants (or σ) as both studies focused on subdiffusion of the loci motion.

Not sure if this comment to the authors or to *eLife*: having only pdf with tracked changes made task of evaluating manuscript unnecessary hard, as reading it and finding right version of figures become cumbersome.

---

## [Author Response]

Essential revisions:Both reviewers believe that the paper has the potential to be published in eLife, but have substantive comments that should be addressed, as detailed below and in their reviews. Please note that both reviewers did not hinge the publication on additional experiments.

We thank the editors and reviewers for assessing our manuscript. The comments and recommendations were useful and constructive and have led to a much improved manuscript. We believe that we have addressed all the concerns raised.

(1) Not all claims are fully supported by the presented data, in particular the claim of the role of ParA hopping/diffusion.

We now clarify that we have not proven that ParA hopping/diffusion is required. Our model simply suggests it. But given how well the model fits the experimental data (varying just 2 parameters), we would argue that it is a strong suggestion.

(2) Limited analysis of the control parameters.

We have now explored how varying system parameters affects the model (new Figure 3—figure supplement 3, Figure 3—figure supplement 4)

(3) Unsatisfactory analysis on the origin of the difference between F1 and pB147 plasmids dynamics.

While we cannot make any new experimental statements on the origin of the difference, we now discuss more clearly what our model suggests. We also add two new supplemental figures to Figure 7.

(4) More careful comparison/analysis to previously published model of ParA and ParA-like systems is an essential element needed to make this work impactful.

We now discuss the existing models in more detail. We added a new section to the methods and a new Table 3 where we compare the ingredients and outputs of the different stochastic models. We discuss the deterministic model of Jindal and Emberly (2019) in some detail in the Discussion section. We also clarify that our model is an extension of the DNA relay model

(5) (Optional) Providing information on the ParA distribution would be a very strong addition.

We now provide example kymographs showing the ParA distribution for the static, diffusive, oscillatory and regular positioning regimes (new panel in Figure 3—figure supplement 2). We leave a more detailed analysis of the ParA distribution and a comparison to experimental measurements for a later study.

Reviewer #1 (Recommendations for the authors):Using "hopping" as a substitute for the ParA diffusion over the chromosome and then stating that it was "primary determinant of the geometry sensing" (abstract) might be misleading. What the authors did – they considered ParA diffusion in the model. And that apparent diffusion over the chromosome might be result of at least two non-exclusive scenarios – repeated cycles of binding/unbinding of ParA dimer intermittent by the diffusion of the unbound ParA dimer or direct hopping of chromosome-bound ParA from one chromosome locus to another when they come into contact upon their intrinsic fluctuations.

Indeed, both scenarios are possible. We have now clarified this and use the term ‘diffusion on the nucleoid’ instead of ‘hopping’ outside of the corresponding section and the methods.

However, we would like to point out that ‘hopping’ has been used in the same context – F plasmid ParA molecules hopping between DNA segments on a DNA carpet (Vecchiarelli et al., 2013).

Along the same line, stating in the Abstract that "we identify ParA hopping on the nucleoid as the primary determinant of this geometry-sensing" is not correct as neither the hopping was explicitly considered in the model nor the author really tested whether this statement would be correct even for ParA diffusion. The only test involved was analysis of the motion in the short cells vs long cells, without perturbation of diffusion per se. Moreover, the authors observed that in case of pB171 plasmid mode of motion was different, yet it is not clear whether the difference could be explained by the model as due to a difference in diffusion coefficient or kd or something else. This reviewer believes that "identify" requires a little bit more that being able to reproduce observed behavior by changing a parameter in the model.

It is true that hopping is not considered explicitly in our model in the sense that we do not explicitly model the transient unbinding events. However, the presence of binding/unbinding events does not change the nature of the equilibrium properties. The equilibrium behaviour is identical to that of free diffusion but with a lower diffusion coefficient. Modelling the cytosolic diffusion of ParA explicitly would slow the simulations significantly and, to our knowledge, has not been done in any of the previous stochastic models.

We agree that ‘identify’ is too strong of a word. We are not able to prove definitively that the lengthscale of ParA hopping is the major determinant for regular positioning. Our model suggests it, as does the result of a length-dependent transition to oscillations. We have re-phrased the sentence in the abstract and introduction. Note also that ParA hopping does have experimental support as we point out (Surovtsev et al., 2016b, Vecchiarelli et al., 2013).

As stated above, we found no fundamental difference in plasmid dynamics in pB171 compared to F plasmid, simply that pB171 has more obvious oscillation in cells with the lowest plasmid concentration. With the understanding of our model, our conclusion is that while pB171 lies somewhat closer to the threshold of the oscillatory instability, both systems are described fundamentally by the same model.

"Lack of quantitative experiments" mentioned several times by the authors might not be exactly the case. While previous experiments/analysis was not the same what the authors did, several groups measured different experimental metrics (just a few examples, far from exhaustive, – Li et al. 2004 Mol.Microbiol, Surovtsev et al. PNAS 2016, Le Gall et al. Nat.Commun. 2016)

This was badly written on our part. Plasmid positions have indeed been quantitatively measured. We were referring to measurements of plasmid dynamics. To our knowledge, the only quantitative measurements of plasmid dynamics are MSD measurements (Figure 3 of Ietswaart et al., Figure S1B of Surovtsev et al. PNAS 2016, Figure S10B of Le Gall et al. Nat.Commun. 2016). We have corrected the phrasing in the abstract and introduction.

The authors compare their model to "diffusive" and "superdiffusive" models (Figure 1 Figure Sup 2A), but details on how they were modeled are lacking.

The three yellow curves are meant as schematics illustrating the qualitative differences between the regimes. The explicit form we use is a standard result and is valid for any process with a MSD that follows a power law t^\α. We now give the explicit form in the legend.

p.5 ln 1-3 The authors report characteristic timescale, τ, at which elastic fluctuations act to be about 120 s using fitting of the velocity vs position data, and then they report that the same τ is ~ 170 s using velocity and position autocorrelation functions fitting, concluding the values are comparable. That seems to this reviewer as quite a difference, warranting at least some comment on the potential origin of the difference.

The position, velocity and autocorrelation data are fit to a model of an overdamped spring. While justified, this is of course an approximation. Hence, we do not necessarily expect the two numbers to be in perfect agreement. Given the 1 minute frame rate of the experiments, we would argue that correlation timescales values of 2 min and 3 min are ‘comparable’. We have now made this clearer in the text and switched from seconds to minutes to make the comparison to the frame rate apparent.

Regarding velocity autocorrelation function and positional dependence, it would be really helpful and reassuring to calculate them from the higher temporal resolution (dt=1s mentioned for MSD and D calculations) since the authors already have the data.

Unfortunately at 1s resolution, the trajectories are not long enough to probe the 2-3 min autocorrelation time due to photobleaching.

For the slope and variance of the velocity profile, the issue, as we discuss in the text, is that at short time scales diffusion dominates, making it more challenging to extract the effective spring constant (the error bars diverge). Conversely, at longer timescales, it becomes challenging to extract the diffusion coefficient. Our 1s data set is also much smaller than our data at 1 min.

p.5 ln 33 …have previously shown that regular positioning can theoretically be achieved, independently of the particular mechanism of force generation, through the balancing of the diffusive fluxes. Given that the force is what really defines where the cargo moves, I don't think the positioning mechanism can be dissected from the mechanism of force generation, once one tries to conclude what specific mechanism operates for a given experimental system. For example, in the model the authors simulated here the force is not directly dependent on the flux of the ParA on the plasmid, rather it depends on the local distribution of ParA. While it is not necessarily negates the authors reasonings, it does require an additional explanation on how these reasonings relates to the simulated model…

While it is correct that the force on the plasmid is due to contacts with nearby ParA dimers, how rapidly those dimers are replaced (and their number) depends on the diffusive flux of dimers into the plasmid. When this diffusive flux is large, which occurs for high \λ, plasmids are regularly positioned. When hydrolysed dimers are not replaced rapidly enough, a depletion zone appears behind the plasmid and processive/oscillatory motion ensues. Thus positioning and force generation can be separated in this sense. While the force is due to a bias in ParA tethers on one side of plasmid versus the other, the presence and nature of the bias is determined primarily by the diffusive flux and the ParA hydrolysis rate. But we agree that of course without the tethers neither occurs.

p.6 Figure 2 (A) When s<<L/2 (i), where L is the nucleoid length, a disparity in the flux only exists very close to the poles (blue region). This seems somewhat counterintuitive, as this regions actually many s away from the sink of ParA (i.e. plasmid)…

We think the following insertion clarifies the sentence: “(A) When *s* ≪ *L*/2 (i), where *L* is the nucleoid length, a disparity in the flux into the plasmid only exists when the plasmid is very close to the poles (blue region).”

When the red-dashed circle (of course in reality it is not a hard threshold) around the plasmid protrudes outside the cell, then the plasmid receives less ParA from that direction. The plasmid locations at which this disparity occurs is indicated by the blue regions (for the long axis which is our interest here).

p.7 30-33 Imaging studies in several Par systems, especially those that position non-DNA cargos, have observed that ParA fluorescence can be higher at the plasmid than elsewhere (Roberts et al., 2012; Schumacher et al., 2017). This is in somewhat disagreement with the canonical picture of the ParB coated cargo acting as a sink for ParA-ATP. This is not a real conundrum, as previous models showed this effect (Surovtsev et al. Biophys.J. 2016, Hu et al.Biophys.J. 2021)

We thank the reviewer for pointing this out. Indeed previous stochastic models have reproduced this behaviour (we were thinking of the deterministic models in which the plasmid is modelled as a sink without a transient plasmid-bound state). We have removed the statement.

Figure 3I What is the color code? It is actually described deep in the methods, but it would be really useful to have it in the main text or figure legend.

We added a short description of the different colors in the legend for Figure 3I.

Figure 3 Sup.Figure -2B what is the color code?

We updated the legend.

In image analysis description, the authors do not provide any details beyond referring to the general Segger description and MotherSegger code on the most important part – cell segmentation and defining position of the plasmid. This reviewer believes that some short description should be readily available within the text for the reader to understand potential limitations. For example, beyond just finding position in the image, how it was used for the analysis – was it position in image coordinate or relative to the cell coordinate, and how change in the coordinate, without motion due to cell growth was taken into account.

We added more details to the description of the pipeline. Briefly, foci positions are found relative to the image coordinate (actually bounding box around the cell) but since cells are vertical in the image (the growth channels are just narrow enough that cells cannot tilt), we transform to the cell coordinate by subtracting half the cell length (for the long axis). The growth between frames is less than 1 pixel (~68 nm) so we neglect its effect (a 2.5 μm cell grows about 17 nm in the 1 min between frames). Accounting for it did not change our results.

It seems that the number of ParA and spring constant values are not specified for the model.

The spring constant of the chromosome fluctuations is related to σ, the standard deviation of the locus position distribution, as k_spring_=1/σ^2. Σ for the two cell axes are specified in the table of parameters (Table 1) and are the same as for the DNA-relay model. The total number of ParA is 500 dimers and is also specified in the table.

Reviewer #2 (Recommendations for the authors):The paper claims that they are the only paper to have a model that shows regular positioning of the ParABS system and that models without substrate hopping on the nucleoid only admit oscillations. This is not true. Jindal and Emberly (2019) showed that regular positioning of plasmids could occur in a model that did not allow for any diffusion of substrate in the nucleoid and that oscillations would emerge due to relaxing of confinement or potentially the liberation of substrate resources due to the addition of plasmids. Indeed the phenomena observed in these experiments (regular positioning, transitioning to oscillations, and back to regular positioning) was predicted in that paper.

We apologise for not discussing this paper. While our focus is on stochastic models, the result that regular positioning can occur in the absence of ParA hopping (i.e. \λ=0) needed to be addressed. We now discuss this in the Discussion section. We believe that the observed effect is the product of a deterministic model. In the Jindal and Emberly model the plasmid interacts with every bound ParA dimers in the cell, albeit with a contribution that is weighted by a Gaussian in the distance. Nevertheless this means that plasmid obtains information from across the cell so that (in the regime where the ParA gradient equilibrates, through un-/binding, faster than the plasmid movement) the plasmid is regularly positioned. In our stochastic model, while it is similarly possible that the plasmid interacts with distant ParA, the finite and low numbers of ParA and the inherent stochasticity of the model, mean that these distant interactions are too rare and noisy to have an effect. When we examined how the phase diagram of our model changes with higher ParA copy numbers, we found only a minor effect on position of the regular positioning regime within the diagram (new Figure 3—figure supplement 3 and Figure 3—figure supplement 4).

Have the authors fully explored the parameter space of their model? If they set kh = 0, (i.e. no hopping), are there any values of n_A, and on/off rates that allow for regular positioning that transitions to oscillations as the cell lengthens? For regular positioning, it requires a broad wake that is balanced between left and right. On longer cells, the confinement is relieved and the complex can oscillate. It would be interesting to know if the stochastic formulation of the model does not allow for any regular positioning if kh=0. If it does, are the parameters values such that they are completely inconsistent with measured kinetic parameters, thus necessitating hopping for the given system.

As we discussed above, it indeed appears that the stochastic nature of the model does not allow regular position to occur without hopping (which is when D_h=0, not k_h=0). We varied the system parameters across 4 orders of magnitude but found no evidence of regular positioning without hopping with one understandable exception (new Figure 3—figure supplement 4). Starting from the static regime, we obtained regular positioning if we increased \σ the length-scale of the chromosome fluctuations by a factor of 10 to about 1 um. This means that each DNA-bound ParA can explore essentially the entire cell (68% of the distribution covers 2um) without hopping and as a result geometry sensing and regular positioning occur. However, this regime is inconsistent with measurements that place the length-scale of the chromosome fluctuations at about 0.1um (Surovtsev et al., 2016).

A similar effect occurs when the number of plasmids is increased. With a sufficiently high density of plasmids, the distance between them becomes comparable to \σ and they compete for the same (non-hopping) ParA dimers. This suppresses the oscillations and leads to regular positioning as then they are equally spaced from each other and the boundaries. This was already seen for the DNA relay model (Surovtsev et al., 2016).

A few other comments/questions:I'm assuming Figure 6 is from experimental data, but there are no reported cell numbers for the various distributions and statistics.

Everything except panel A is from experimental data. We have now added the number of cell cycles analysed to Figures 6 and 7.

it would have been nice to have seen data from > 2 plasmids. Do the authors ever see oscillations in 2 plasmids switching to regular positioning once 3 plasmids are present (i.e. Figure 7F with a column for 3 plasmids). Presumably yes as there are around ~20% of the 2 plasmid systems oscillating, and when 3 are present, regular positioning likely follows. Do they ever get filamentous cells, and what are the dynamics like in those cells?

The labelling system we use for pB171 gives a reasonably high background and broad foci. As a result we are not confident in analysing plasmid trajectories in cells with more than 2 plasmids. However, we have now analysed the 1->2 plasmid transition (Figure 7F concerns the frequency of oscillations in the population before/after transitions), which we discuss below. Regarding filamentous cells, we very rarely get them. They have regular-spaced plasmids across some fraction of the cell, presumably coinciding with the nucleoid. However, we do not have enough of these cells to make quantitative statements about the dynamics.

I am intrigued by the difference in dynamics for the F-plasmid and pB171 plasmid. Their experimental results for the 1b system show it is more likely to oscillate. Why? Is it due to a smaller s? The paper claims that it is due to smaller s, but no real discussion/evidence is given.

Indeed, we do not have evidence that s, or rather λ, is smaller for pB171. Our model indicates that it is closer to the threshold of the oscillatory instability (crossing fully into oscillations for the subpopulation of cells with a single plasmid). We placed pB171 in the phase diagram in Figure 8A with a lower λ but this was somewhat arbitrary and not properly discussed. As discussed in the public comments, we have now clarified this.

I could find no details of how varying n_A affects results. As in most other published models, this also has a huge effect on dynamics, similar to their parameter, λ.

As we discussed above, varying n_A had little effect on the fundamental nature of the dynamics though did of course modulate the degree of stochasticity (new Figure 3—figure supplements 3 and 4). It is possible that for very high n_A, we approach the deterministic model of Jindal and Emberly (2019) and the regular positioning regime extends to λ=0. However, at realistic levels of hundreds to a few thousand ParA (we go up to 1000 dimers in Figure 3—figure supplement 3 and up to 50000 in Figure 3—figure supplement 4), we see little effect.

Could some of their observations be due to cell-to-cell heterogeneities in n_A? Also dilution would have an effect, which it is not clear if it is taken into account here.

As mentioned, n_A appears to have little effect on the nature of the dynamics. Still, it is true that ParA is inherited asymmetrically by daughters cells (Hu et al. 2021, Biophysical Journal). However, since plasmids are extremely stable within cells, this suggests that asymmetry in ParA levels at birth resolves within one cell cycle. Otherwise, a subpopulation would lose ParA and plasmid loss would ensue. This also connects to the question below regarding correlations between generations.

On the other hand, plasmid replication does have an effect on oscillations, consistent with our model. We found (new FIgure 6—figure supplement 2) that oscillations (as measured by the sign of the velocity autocorrelation at lag 1) disappeared upon plasmid replication (foci splitting), while ParA levels would not be expected to change much during this time.

Do they use n_A=500 for all simulated cell lengths? Could differences in the total amount of ParA explain the different dynamics between the F plasmid and pB171 (see my comment above)?

For simplicity, we use n_A=500 for all domain lengths but n_A would not be expected to vary much between the two standard lengths we use (L=2.53 and 2.91 – the mean length of cells carrying one or two F plasmids respectively, a 15% difference). In any case, as mentioned above n_A did not have a substantial effect on the nature of the dynamics so it seems unlikely to explain the difference between F plasmid and pB171.

Have they done lineage tracking? Do they see correlations in the likelihood to do regular positioning or oscillations? If so, especially for the case with 1-plasmid oscillations, is it due to length differences in the daughter? or could oscillations be arising from some other unmeasured system parameter?

We have now performed such a tracking analysis. We examined if oscillations were inherited or if they only arose in a related subpopulation of cells (e.g. cells from the same few channels). We found that oscillations occurred randomly in space (microfluidic channel) and time and found no evidence of hereditary oscillations beyond that expected from the decorrelation time of the plasmid concentration (1-2 generations) (new Figure 6—figure supplement 3).

[Editors’ note: further revisions were suggested prior to acceptance, as described below.]

Reviewer #2 (Recommendations for the authors):Suggestions/Questions:While the revised manuscript now really helps the reader to understand how ParA/ParB system works, thanks to explicit comparison to earlier models, here are a few things that could be addressed by the authors (in this reviewer opinion).Again, there is no doubt that λ is an important parameter of the model, however I found the authors explanation (Figure 2) confusing (at least for me). They argue about importance of the parameter based on the importance of the balancing ParA fluxes to the plasmid. But these fluxes would be there and would be balanced only in the center (for one plasmid) no matter how big a nucleoid is relative to the ParA diffusion scale…

This is in principle correct. However the difference in the incoming fluxes becomes infinitesimal over the interior of the domain as λ decreases (The spatial dependence looks like the velocity profile in Figure 3C,D,E., rather than that of Fig, 3H. See also Murray and Howard, 2019 and Subramanian and Murray, 2021). Clearly, in a stochastic model, at some point this will not be sufficient to produce regular positioning. Even in a continuous model, other effects can begin to play a greater role. This has been seen in the model of Walter et al. 2017, in which regular positioning transitions to oscillations as the diffusive lengthscale is decreased (I don’t say λ because in this model the finite diffusive lengthscale is of the cytosolic state not the nucleoid bound state, as we discuss in the Discussion section, but the result is the same.)

We now clarify this point in the text.

Moreover, the plasmid interacts with ParA bound mostly within few σ (range of chromosome fluctuations), so argument about "information" also does not work out…

As discussed in the text, Figure 2 and mentioned above in the previous comment, it is the disparity in the fluxes of ParA into the plasmid that provides the positional information. Tethers break due to hydrolysis and must be replaced. If they are replaced faster on one side of the plasmid than the other, then this results in a net force acting on the plasmid.

Also, the authors did not really test whether λ (but not Dh alone) governs dynamics. The authors varied λ and epsilon independently by changing Dh and kh, but does plasmid dynamics look exactly the same if we say change instead Dh, kd, kh and L such that λ and epsilon do not change?

We have now added an additional Figure (Figure 3—figure supplement 5) in which we change Dh, kd, kh simultaneously by the same factor such that neither λ nor epsilon changes. We also scale ka in the same way so that the number of bound ParA in the absence of a plasmid (as measured by theta) is constant.

This sweep supports our conclusion on the importance of λ and epsilon. We saw that regular positioning was maintained over 2 orders of magnitude, with the only change being the frequency of the noise. This is expected since changing these parameters as we did, modulates the rate at which plasmid-ParA tethers turnover (and hence the timescale of fluctuations in plasmid position), while leaving λ, epsilon and theta fixed. The same behaviour was found in the static and diffusive regimes. In the oscillatory regime, we found that the period of oscillations increased inversely to the changed parameter values consistent with longer-lived tethers resulting in slower motion. However, for faster tether turnover, the dynamics become noise-dominated since the few tethers present in this regime are too short-lived to pull the plasmid and maintain directed motion (the timescale of the tether pulling depends on σ_x,y and D_p which are not changed in the sweep).

We did not vary L as suggested as this would change several other system parameters such as the ParA density, the relative area occupied by the plasmid, etc. We also have precise measurements of cell length and deviating too far from these measurements has limited biological relevance.

Other parameters sweeps that were added to the manuscript are very appreciated, but they do not answer this question. Along the same line, a bit more expanded discussion on underlying nature of the transition to different dynamics during these sweeps may help reader to understand interplay between different parameters in determining dynamics qualitatively. I found description of results of these sweep too brief (so not very insightful).

We are unclear what this reviewer means by ‘nature’. While the transition to oscillations can likely be identified as some kind of Hopf bifurcation, a stability analysis of our multi-parameter stochastic model is beyond the scope of this manuscript.

The reviewer may be referring to an intuitive/physical explanation. To that end, we have now added a discussion of the transitions seen in the sweeps to the legend of Figure 3—figure supplement 4.

Along the same line, it might be a bit counter-intuitive that the system behavior almost does not depend on number of ParA. For example, the authors argued for the importance of λ based on ParA-fluxes, but value of the fluxes should strongly depend on the amount of ParA in the system.

We could have been clearer about this. Of course, without ParA there can be no non-trivial dynamics. While the boundaries between the regions in the phase diagram stay roughly unchanged, all the regimes become progressively more noisy as the amount of ParA is decreased until diffusion dominates i,e, all regions transition to diffusion. We now show this explicitly by adding sweeps for n_A=5 and 10 to Figure 3—figure supplement 3. Our explanation for the similar dynamics at high ParA amounts is due to the fact that in our model existing tethers need to disassociate to allow further movement of the plasmid. Therefore, increasing the number of ParA has little effect since the tether hydrolysis rate limits plasmid movement.

Additionally, the authors report that the plasmid velocity strongly depends on the ParA amount associated with it (Figure 3—figure supplement 2 B) (perhaps, on overall amount of ParA as well). One might think that the velocity would play a role whether we observe a strongly dampened spring or a decaying oscillator. Maybe it is a naïve thinking, but, perhaps, this warrants some explanation in the manuscript.

The reviewer is correct. As stated in the legend the colour of the data points is the same as that in Figure 3, indicating that oscillations are associated with the highest plasmid velocities.

Note that we do not discuss decaying oscillations. The oscillations of our model are stable.

And, while for relatively high ParA number the dependence might be saturating, Figure 3—figure supplement 3 top-left and Figure 3—figure supplement 4D suggest that lowering ParA may drive switch to different dynamics.

See above. The system becomes diffusive once there are insufficient ParA tethers to quench the intrinsic diffusion of the plasmid.

Additional comment on parameter sweeping. Since ParA diffusion is an "effective" description of some underlaying dynamics, effective Dh might depend on other parameters, i.e. cannot be varied independently. For example, in a potential "binding-unbinding-bulk_diffusion-binding" scenario, Dh depends on k_dis, ka and D_bulk. In an alternative scenario, where ParA hopes (without unbinding) to a new DNA position, Dh depends on σ and Da. While ideally these scenarios should be modelled explicitly to test how changes in these parameters affect apparent Dh and plasmid dynamics, such a limitation, perhaps, should be mentioned in the manuscript (since it will make the model more complex and also there is only so much we can do at once).

We now acknowledge that in the alternative scenario, Dh would indeed depend on σ and Da.

Regarding pB171 plasmids, having a plot similar to Figure 1E and F would be nice, as they were used as an evidence for 'true positioning' regime.

The velocity profile was only one piece of evidence. As seen by comparing Figure 3G and H, the profile can look similar even for very different dynamics (especially given our much smaller pB171 data set). This is why we focus on the PAC and VAC, which are fundamentally different in the two regimes (Figure 3—figure supplement 1).

And still some potential explanation of what might be different between F and pB171 plasmids – D, kd(?) – would be a nice addition as it might prompt someone to test it in the experiment.

We proposed that λ is smaller for pB171 than for F plasmid. We have no evidence or other knowledge to say anything further. Either or both the hydrolysis rate or the DNA un/binding rates (hence Dh) could be different. Their protein sequences are substantially different and they come from different ParA families.

On presentation:I found the title of the paper too vague, as "nature of plasmid positioning" could be interpreted very differently (and thus whether the work "uncovers" it or not).

We disagree with this and do not see the different interpretations. The nature of the dynamics (regular positioning, oscillating or something else) of the two most well-studied plasmid-based ParABS systems, F and pB171, was unclear. Our work has resolved this.

Furthermore, neither of the two other reviewers raised an issue with the title. We therefore respectfully request the title to remain as it is.

I would suggest adding to the abstract what were the key ingredients of the authors model to succeed in achieving full range and transition between different modes of the plasmid dynamics.

In the initial submission, we had included our identification of “ParA hopping” as the determinant of the dynamics but removed it at Reviewer 1’s suggestion as we have not explicitly proved that hopping is actually required. We believe the importance of this point depends on the background/interest of a reader. We would like our work to be read by both experimentalists and theorists and we think leaving the abstract as it is strikes the right balance.

Still in this reviewer opinion, "lack of quantitative measurements of plasmid dynamics" (used several times through the paper) might be misleading as the authors measured from microscopy exactly the same thing – position vs time – as earlier works. The strength of the work is not in a measuring experimentally a new thing, but having a great statistics (high-throughput imaging!) that enabled a new analysis (meaningful beyond inherent stochastic noise) – velocity vs position and velocity and position autocorrelation functions. I would emphasize this achievement instead.

As we explained, other studies generally rely on snapshots and/or only show a few timelapses for illustrative purposes and not for analysis (note that what look like kymographs in some papers are actually demographs i.e. built up from snapshots not timelapses). As we stated in our previous response, the only other quantitative dynamic measurements that we are aware of are the MSD curves of a few studies which are produced by averaging tracks from many cells and consider only displacements and not the position of those displacements. Thus previous position vs time measurements/analysis have been limited.

While our high-throughput and data analysis approaches are important parts of the paper, the work would be much less impactful without our ‘unifying’ model (as noted by reviewer 1). We believe that both parts are equally important and have tried to give a balanced presentation.

Figure 1E referred before any other panels, and similar happens with some other panels through the manuscript.

Panels A, B are cited in the first paragraph of the Results section. Panels C and D are cited in the second paragraph.

Table1: I do not think Weber et al. 2010, or Javer et al. 2014 reported any chromosomal spring constants (or σ) as both studies focused on subdiffusion of the loci motion.

The chromosomal spring constant was estimated (not taken) from these papers by Surovtsev et al., 2016. We simply wanted to give the original references cited by Surovtsev et al. rather than just refer to their paper. We now cite only Surovtsev et al., 2016.

Not sure if this comment to the authors or to eLife: having only pdf with tracked changes made task of evaluating manuscript unnecessary hard, as reading it and finding right version of figures become cumbersome.

We have tried to tidy up the tracking of figures in the tracked Word files.